# Predicting Quality of Life in Relapsing–Remitting Multiple Sclerosis: Clinical Burden Meets Emotional Balance in Early Disease

**DOI:** 10.3390/neurolint17120195

**Published:** 2025-12-01

**Authors:** Emilio Rubén Pego Pérez, María Lourdes Bermello López, Eva Gómez Fernández, María del Rosario Marín Arnés, Mercedes Fernández Vázquez, María Irene Núñez Hernández, Emilio Gutiérrez García

**Affiliations:** 1Department of Psychiatry, Radiology, Public Health, Nursing and Medicine, University of Santiago de Compostela, Avda. Xoán XXIII s/n, 15704 Santiago de Compostela, Spain; 2Faculty of Nursing, Avda. Xoán XXIII s/n, 15704 Santiago de Compostela, Spain; 3Research Group CroniEduTec (Chronicity, Educational Innovation and Applied Health Technologies), Faculty of Nursing, University of Santiago de Compostela, 15704 Santiago de Compostela, Spain; 4Nurses of Neurology and Neurosurgery of Lucus Augusti Hospital, 27004 Lugo, Spain; lubermello@hotmail.com (M.L.B.L.); buchi@hotmail.es (E.G.F.); charochili@yahoo.es (M.d.R.M.A.); mercedes.fernandez.vazquez@sergas.es (M.F.V.); irene.nunez.hernandez@sergas.es (M.I.N.H.); 5Department of Clinical Psychology and Psychobiology, Research Group: GIBIOMED Group of Biomedical Research, Faculty of Psychology, University of Santiago de Compostela, 15703 Santiago de Compostela, Spain; emilio.gutierrez@usc.es

**Keywords:** relapsing-remitting multiple sclerosis, quality of life, health assessment, follow-up, nursing, autoimmune diseases, health status indicators, social interaction, psychological distress, emotional exhaustion

## Abstract

Background/Objectives: Relapsing-remitting multiple sclerosis (RRMS) is a chronic neurological disease that significantly impacts health-related quality of life (HRQoL). This study aimed to analyze the evolution of HRQoL in individuals with RRMS, identify associated factors, and determine predictive variables. Methods: A prospective observational study was conducted with 35 participants diagnosed with RRMS at the Lucus Augusti University Hospital between January 2023 and March 2025. HRQoL was assessed using the MSQOL-54 questionnaire at baseline, 3 months, and 6 months. Data were analyzed using non-parametric tests to account for the small sample size and non-normal distribution of the variables. Results: Results showed overall stability in HRQoL (mean score: 62.4 ± 14.1 at baseline, 62.8 ± 12.7 at 3 months, and 62.4 ± 11.8 at 6 months), although significant declines were observed in emotional limitations (64.4 ± 23.0 at baseline to 58.9 ± 20.5 at 6 months) and social functioning (70.5 ± 16.7 at baseline to 65.5 ± 12.8 at 6 months). Improvements were noted in pain perception (78.9 ± 23.6 at baseline to 81.8 ± 20.5 at 6 months) and stress (44.3 ± 22.5 at baseline to 48.9 ± 17.8 at 6 months). Factors such as family history (associated with mental health at diagnosis, *p* = 0.028), autoimmune diseases (associated with physical function at diagnosis, *p* = 0.035), and lifestyle habits (e.g., tobacco use associated with physical limitations at 3 months, *p* = 0.045) were significantly associated with HRQoL. Baseline HRQoL emerged as a strong predictor of future scores (Spearman’s correlations, *p* < 0.01), indicating that early assessments may guide interventions. Conclusions: Although overall HRQoL remains stable in RRMS, specific domains such as emotional and social functioning exhibit progressive decline, highlighting the need for tailored interventions. The findings underscore the importance of integrating early psychosocial support and lifestyle interventions into routine care to mitigate vulnerabilities in emotional and social domains of HRQoL.

## 1. Background

Multiple sclerosis (MS) is a chronic, demyelinating, autoimmune, inflammatory, and degenerative neurological disease that affects the central nervous system, particularly the brain and spinal cord. Despite extensive research, its etiology remains undetermined, with evidence pointing to multifactorial causes, including genetic, immunological, and environmental factors [1,2,3,4]. The interplay between these factors highlights the complexity of MS, making it not only a challenge for scientific understanding but also a significant burden for healthcare systems worldwide [5,6]. Advances in research have provided valuable insights into the mechanisms underlying MS, yet its unpredictable progression continues to pose difficulties for effective management and long-term care [7,8].

The clinical manifestations of MS are diverse and debilitating, encompassing motor, sensory, visual, urinary, and cognitive impairments, as well as neuropsychiatric symptoms such as anxiety and depression. These symptoms, which vary greatly among individuals, can affect virtually all aspects of daily life, often leading to a profound loss of independence and quality of life (QoL) [9,10,11]. They interfere with treatment adherence and limiting their ability to cope with daily challenges. Anxiety and depression, highly prevalent in people with MS, are further exacerbated by social isolation and stigma, which reduce community support and amplify emotional distress. Emerging evidence suggests that early interventions targeting emotional and social dimensions can significantly improve long-term outcomes in RRMS patients [12,13,14]. Stigma, in particular, has been shown to hinder open communication about the disease, further isolating individuals and limiting their access to necessary resources. However, robust social networks have been shown to act as protective factors, enhancing health perception and the ability to manage relapses and physical limitations [2,15,16]. The role of social support in mitigating the emotional and physical impact of MS underscores the importance of fostering inclusive environments and community-based interventions [17,18,19].

Globally, MS impacts approximately 2.8 million individuals, with over 55,000 diagnosed cases reported in Spain (80 to 180 cases per 100,000 inhabitants) [20]. This increase in prevalence is not only attributed to improved diagnostic capabilities but also to greater awareness and reporting of the disease, particularly in regions with historically lower detection rates [21,22,23]. In Galicia, prevalence rates range between 140 and 183 cases per 100,000 inhabitants, with an average age of onset of 29 years [20]. The disease predominantly affects women aged 20 to 40 years, with a female-to-male ratio of approximately 2:1. The prevalence of MS is also higher in regions at elevated latitudes [3,4,24,25]. This geographical disparity has been linked to environmental factors such as reduced sunlight exposure, which may contribute to vitamin D deficiency—a potential risk factor for MS [26,27]. Furthermore, viral infections and other immune-modulating factors are believed to play a critical role in the disease’s etiology, further emphasizing the multifactorial nature of MS [27,28,29].

Clinically, MS is categorized into four primary forms: relapsing-remitting MS (RRMS), secondary progressive MS, primary progressive MS, and progressive-relapsing MS. Among these, RRMS is the most common, accounting for approximately 85% of diagnoses [2,30]. RRMS is characterized by episodes of neurological dysfunction or relapses, followed by periods of remission during which symptoms stabilize. To establish the diagnosis of RRMS, the 2017 McDonald criteria were utilized, which emphasize the requirement for dissemination in both time and space of central nervous system lesions confirmed through clinical and radiological assessments [1]. These criteria were selected for their high sensitivity and specificity, facilitating early and accurate diagnosis.

In addition, the MSQOL-54 instrument was chosen to evaluate health-related quality of life (HRQoL) due to its disease-specific design and multidimensional approach. Unlike generic QoL instruments, the MSQOL-54 incorporates domains particularly relevant to MS, such as fatigue, cognitive function, and social support, which are critical for understanding the unique challenges faced by individuals with this condition. This selection aligns with the recommendations outlined in the “Proceso Asistencial Integrado para el abordaje de la esclerosis múltiple en Galicia,” which underscores the importance of using validated tools tailored to the MS population to ensure comprehensive and patient-centered care [31,32,33]. During relapses, individuals may experience optic neuritis, sensory disturbances, motor dysfunction, sphincter issues, spasticity, and cognitive impairments, predominantly caused by immune cell migration to the central nervous system, leading to inflammation [2,15,16]. Understanding the mechanisms driving these relapses remains a critical focus of research, as it holds the potential to improve therapeutic strategies and reduce the burden of disease progression [34,35,36].

The socioeconomic burden of MS is substantial, with direct costs related to treatment and follow-up ranging from €10,486 to €27,217 per person annually, depending on the degree of disability. These costs are further compounded by non-healthcare expenses, such as transportation, home adaptations, informal care, and work absences, which can amount to €454 to €25,850 annually [25,37]. These figures highlight the significant financial strain placed on individuals and families, particularly those with limited resources or unstable employment [38,39,40]. Individuals with lower incomes or precarious employment face greater barriers to accessing healthcare services, exacerbating the emotional and social impact of MS [25,37]. This economic disparity underscores the need for policies aimed at improving access to affordable care and support services for vulnerable populations [41,42].

Health-related quality of life (HRQoL) has emerged as a key indicator for evaluating the comprehensive care of individuals with MS. It encompasses physical, emotional, social, and functional dimensions, including fatigue, physical disability, sleep disorders, and depressive mood, which collectively limit autonomy, interpersonal relationships, and work capacity [20]. Fatigue, one of the most pervasive symptoms of MS, often goes underrecognized despite its profound impact on HRQoL. Similarly, sleep disturbances and depression frequently coexist, creating a cycle of worsening symptoms that further deteriorates overall well-being [7,43,44,45]. The variable course of MS, coupled with the frequent emergence of psychological comorbidities, further increases the direct and indirect costs for both individuals and healthcare systems [25,37]. HRQoL provides a patient-centered perspective, allowing healthcare providers to address not only the physical symptoms of MS but also the broader psychosocial challenges faced by individuals. This holistic approach is essential for developing interventions that not only target clinical outcomes but also improve patients’ overall QoL [46,47,48,49].

Measuring HRQoL provides a more holistic understanding of the challenges faced by individuals with MS, facilitating a person-centered approach to care. Key factors such as fatigue, sleep disorders, physical disability, and depressive mood significantly deteriorate HRQoL, making its assessment essential for evaluating treatment effectiveness and identifying psychosocial factors that require intervention [50]. Furthermore, understanding HRQoL can aid in the development of targeted interventions aimed at improving not just clinical outcomes but also the overall well-being of individuals with MS. By focusing on HRQoL, healthcare providers can better tailor treatment plans to address the unique needs of each patient, ultimately enhancing their ability to manage the disease and maintain a sense of normalcy in their lives [45,46,48].

Despite the growing recognition of HRQoL as a critical outcome in MS care, there is limited research analyzing its evolution in individuals with relapsing-remitting MS who have not undergone specific interventions. Most existing studies focus on the effects of pharmacological treatments or rehabilitation programs, leaving a gap in understanding the natural progression of HRQoL in untreated populations. This lack of data highlights the need for longitudinal studies that monitor HRQoL over time, providing insights into its trajectory and identifying opportunities for early intervention [7,43,45,46,48]. This study aims to address this gap by examining the QoL in individuals with RRMS, focusing on the influence of sociodemographic and clinical variables.

The main objective of this study is to analyze the quality of life in individuals diagnosed with RRMS over three and six months follow-up based on sociodemographic and clinical variables. The specific objectives are as follows: (1) To describe the sociodemographic, clinical, and quality of life characteristics of the study population; (2) To determine the relationship between sociodemographic and clinical variables and quality of life, and (3) To identify predictive factors for quality of life.

By addressing these objectives, this study seeks to contribute to the understanding of how RRMS affects QoL and to provide insights that may inform future interventions aimed at improving outcomes for this population. Ultimately, the findings aim to bridge existing gaps in literature and support the development of holistic care models that prioritize the needs and experiences of individuals with RRMS.

## 2. Methods

### 2.1. Study Type

This study follows an observational, analytical, and prospective design, conducted on users treated at the Neurology and Neurosurgery Unit or the neurology nursing clinic of the Lucus Augusti University Hospital (HULA), diagnosed with RRMS between January 2023 and March 2025.

### 2.2. Population, Sample, and Inclusion Criteria

The target population for this study consists of individuals diagnosed for the first time with RRMS and attended by physicians from the Neurology and Neurosurgery Service at HULA. HULA is located in the city of Lugo, within the same province, which includes 332,100 inhabitants registered with a healthcare card as of February 2017.

Regarding the sampling strategy, the study recruited all available patients diagnosed with RRMS during the inclusion period, thus encompassing the entire eligible population. While randomization or stratification was not applicable in this context due to the nature of the cohort, additional literature has been incorporated to strengthen the methodological framework and align with best practices in observational studies.

This study follows a longitudinal design with a two-year time window, during which patients were included according to their diagnosis and followed up at baseline, 3 months, and 6 months.

QoL questionnaires were administered at the time of initial diagnosis and at 3 and 6 months of follow-up. The sample consisted of a convenience group of individuals with RRMS selected consecutively for this study.

The inclusion criteria were individuals aged 18 years or older, residing in Galicia, and diagnosed with RRMS according to the 2017 McDonald criteria, which require evidence of dissemination in space (lesions in at least two of four CNS regions: periventricular, cortical/juxtacortical, infratentorial, or spinal cord) and time (simultaneous presence of enhancing and non-enhancing lesions or new T2/enhancing lesions on follow-up MRI). Participants had to provide informed consent and undergo regular clinical follow-up within the framework of the Integrated Care Process for MS in Galicia. Exclusion criteria included individuals with other MS phenotypes (e.g., primary or secondary progressive MS), severe comorbidities affecting HRQoL evaluation, pregnancy or lactation, refusal to participate, withdrawal of consent before data collection completion, or loss to follow-up.

### 2.3. Justification and Sample Size Calculation

RRMS diagnoses account for approximately 80% of all individuals with MS. MS affects approximately 0.1% of the population. The proportion (P) was set at 0.8, based on epidemiological data indicating that RRMS represents 80% of all MS cases. Therefore, in the Lugo healthcare area, there would be a total of 332 individuals with MS, of whom 265 would have RRMS.

A 90% confidence interval was selected to balance precision and feasibility given the pilot nature of the study and the operational constraints of the healthcare service. A 14% margin of error was chosen to reflect the variability expected in the population, considering the small sample size and the exploratory nature of the study. The formula used for the calculation was: $$\mathrm{Sample}\;\mathrm{Size}\;=\frac{\frac{z^{2}*P(1-P)}{e^{2}}}{1+(\frac{z^{2}*P\left(1-P\right)}{e^{2}N})}$$

The study employed convenience sampling, selecting accessible and available individuals from the Neurology and Neurosurgery Unit and the Neurology Clinic at HULA, which explains the high participation rate (>98%). However, the final sample included 35 users due to logistical and temporal constraints. This reduction is justified by the duration of the inclusion period, the availability of individuals, and operational limitations of the service during this pilot phase.

The sample size aligns with similar studies in RRMS populations, considering the low prevalence of the disease and the specific inclusion criteria. Non-parametric tests ensured robust statistical analysis despite the limited sample size.

### 2.4. Variables

#### 2.4.1. Independent Variables

Socio-epidemiological factors: sex, age, ethnicity, education level, marital status, employment status, and annual income.Clinical factors: presence of family history, autoimmune diseases, previous mononucleosis, pregnancy planning, tobacco and alcohol consumption, ongoing treatment, and initial symptoms.

#### 2.4.2. Dependent Variables

Quality of Life: physical health, role limitations related to physical or psychological problems, pain, mood, energy, health perception, social functioning, cognitive functioning, health-related concerns, and overall perception of QoL.

### 2.5. Instrument

The Multiple Sclerosis Quality of Life-54 (MSQOL-54) is a specific tool designed to assess QoL in individuals with MS. This questionnaire combines 18 MS-specific items, developed through expert opinions and a literature review, with 36 items from the Short-Form 36-item (SF-36), a widely used generic instrument for evaluating QoL. In total, the MSQOL-54 consists of 54 items organized into two main scales: Physical Health and Mental Health, which are further divided into 12 multi-item subscales (Physical Function; Physical Limitations; Emotional Limitations; Pain; Mental Health; Energy; Health Perception; Social Function; Cognitive Function; Stress; Sexual Function; and Overall Perceived Quality of Life) and two single-item subscales (Health Changes and Satisfaction with Sexual Life).

Each subscale has a scoring range from 0 to 100, where values closer to 0 indicate poorer perceived QoL, and higher values reflect better perceptions of QoL. Subscale scores are calculated by summing the corresponding item values and transforming the result into a 0–100 scale using a proportional conversion formula. To facilitate interpretation, quantitative scores are categorized qualitatively into five levels: Extreme impairment (0–24), Quite impaired (25–49), Moderately impaired (50–74), Slightly impaired (75–99), and not impaired (100).

The MSQOL-54 has demonstrated strong validity in terms of content, constructs, reliability, discrimination, and responsiveness. Additionally, it exhibits high internal consistency, with Cronbach’s alpha values ranging from 0.75 to 0.96, supporting its reliability as a tool for assessing QoL in individuals with MS [20,51].

The instrument was validated in Spanish, with dimensions showing high internal consistency (Cronbach’s alpha: 0.70 to 0.92) across all areas except for two domains. Regarding external validity, a significant correlation was observed between the global index and all dimensions (Pearson coefficients: 0.46 to 0.76). The instrument’s reproducibility was satisfactory (intraclass correlation coefficients: 0.60 to 0.91). The questionnaire’s acceptability was high, and the average time required for completion was 9.8 ± 11.8 min [52].

This instrument provides a comprehensive view of perceived QoL, encompassing physical, emotional, and social aspects, and facilitates the identification of specific areas requiring intervention to improve the well-being of individuals.

### 2.6. Data Collection

The data gathered for this study were completed by the principal investigator and were only accessible to them and the project’s collaborative team. No external individuals outside the research team were permitted to make changes to the data. Data were collected at three specific points: at the initial diagnosis of users (in the Neurology Unit or Clinic), at three months, and at six months (in the Neurology Nursing Clinic).

For the evaluation, the MSQOL-54 was used to assess QoL and its dimensions. The instrument was administered by trained nurses, and the data were recorded in a specifically designed collection notebook for this study. The information was stored anonymously and reviewed to ensure its quality and consistency.

To ensure methodological rigor and consistency in data collection, all data collectors received specific training at the Faculty of Psychology of the University of Santiago de Compostela. The research team was thoroughly instructed on the use and application of the MSQOL-54 instrument, including its scoring system and interpretation. Training sessions included both theoretical instruction and practical exercises, with trial runs conducted to ensure uniformity in administration and data recording.

#### Handling of Missing Data

No missing data were recorded in this study. All participants completed the data collection process in full, ensuring a complete dataset for analysis. This was achieved by implementing rigorous data monitoring procedures and maintaining close communication with participants throughout the study period.

### 2.7. Data Confidentiality and Ethical Considerations

This study received approval from the Santiago-Lugo Research Ethics Committee (Registration Code: 2022_388).

The project was conducted in strict adherence to the ethical principles outlined in the Declaration of Helsinki by the World Medical Association (2024), as well as the current legal regulations in Spain (Organic Law 3/2018 on Personal Data Protection and Guarantee of Digital Rights, Law 41/2002 on Patient Autonomy, and Law 3/2005 regarding access to electronic medical records).

To ensure participant privacy protection, clinical data were encoded and dissociated, guaranteeing that no identifiable information was included in the database. Only the principal investigator had the ability to associate data with specific individuals, and the information collected was managed to maintain anonymity. Upon study completion, the data will either be destroyed or retained in anonymized format, as stipulated in the informed consent signed by participants. HULA is the center responsible for data processing.

### 2.8. Data Analysis

A descriptive analysis was performed using measures of central tendency, such as the mean (M), and measures of dispersion, represented by the standard deviation (SD), for quantitative variables. For qualitative variables, absolute frequencies and percentages were used.

The Shapiro–Wilk test was applied to evaluate normality. Results indicated that all quantitative variables, except income, had a non-parametric distribution. Consequently, non-parametric tests were employed for all analyses, including cases where a variable with normal distribution was correlated with others exhibiting non-parametric distributions.

The statistical tests utilized included the Kruskal–Wallis test for comparisons across more than two groups, the Mann–Whitney U test for pairwise difference analysis, and Spearman’s correlation coefficient to examine the strength and direction of associations between quantitative variables.

All analyses were conducted using PASW statistical software (version 23.0; SPSS Inc., Chicago, IL, USA), and a bilateral significance level of *p* < 0.05 was considered.

## 3. Results

### 3.1. Sociodemographic Characteristics

#### 3.1.1. Age

The sample consisted of 35 individuals, with a mean age of 38.29 ± 10.38 years, ranging from 20 to 59 years (Table 1). After categorizing this variable, the age groups were distributed as follows: the group of individuals under 28 years represented 20% of the sample, the group aged 29–38 years accounted for 34.3%, and the group over 39 years comprised 45.7%. The predominant group was participants aged over 39 years.

A significant relationship was found between age and QoL at diagnosis and three months (*p* = 0.018 and *p* = 0.024, respectively), but no significant association was observed at six months (Figure 1). Differences were evident when comparing age groups in terms of QoL at diagnosis and at three months (χ^2^ = 8.789; df = 2; *p* = 0.012 and χ^2^ = 8.415; df = 2; *p* = 0.015, respectively). The mean ranks for QoL at diagnosis were 26 for the ≤28 years group, 13.33 for the 29–38 years group, and 18 for the ≥39 years group. At three months, the mean ranks were 26.50 for ≤28 years, 14.88 for 29–38 years, and 16.63 for ≥39 years.

#### 3.1.2. Sex

Regarding sex, 57.1% of participants were women, while 42.9% were men. No significant differences were observed between groups.

Income, Marital Status, Education Level, and Employment.

Annual income distribution revealed that 40% of participants had incomes below €12,450, 31.4% between €20,200 and €35,200, 5.7% between €35,200 and €60,000, and 2.9% over €60,000. No individuals reported incomes exceeding €300,000.

Marital status distribution showed that 54.3% of participants were married, 40% were single, and 5.7% were in a domestic partnership. Regarding education level, 40% of participants had university studies, 22.9% completed secondary education (ESO), 14.3% had higher vocational training, 11.4% completed high school, and 11.4% had intermediate vocational training.

Marital status showed interactions with the stress dimension at diagnosis (*p* = 0.034).

Education level correlated with the cognitive function dimension at diagnosis (*p* = 0.04) and at three months (*p* = 0.025).

Employment status showed significant correlations with the energy dimension at diagnosis (*p* = 0.024), at three months (*p* = 0.039), and at six months (*p* = 0.001). It also interacted with the social function dimension at three months (*p* = 0.012) and six months (*p* = 0.001), as well as with the stress dimension at three months and six months (*p* = 0.001 for both). Finally, employment status revealed relationships with the emotional limitation dimension (*p* = 0.007) and health changes dimension (*p* = 0.012) at six months.

### 3.2. Clinical Characteristics

#### 3.2.1. Family History and Previous Diseases

Eighty percent of participants reported no family history, while 20% did. Regarding autoimmune diseases, 82.9% did not have such conditions, while 17.1% reported them. Additionally, 94.3% had no history of previous mononucleosis. Most participants were employed (42.9%), followed by self-employed individuals (31.4%) and students (22.9%).

A relationship was identified between family history and the mental health dimension at diagnosis (*p* = 0.028), as well as with the health changes dimension at three months (*p* = 0.025). Finally, family history was associated with the energy dimension at six months (*p* = 0.008).

A significant relationship was found between autoimmune diseases and physical function at diagnosis (*p* = 0.035). Associations were also observed with the stress dimension (*p* = 0.031) and health changes dimension (*p* = 0.002) at three months, and with the pain dimension at six months (*p* = 0.038).

Previous mononucleosis was related to the social function dimension at diagnosis (*p* = 0.043) and the sexual function dimension (*p* = 0.037).

Pregnancy planning was related to the sexual satisfaction dimension (*p* = 0.02) and the sexual function dimension (*p* = 0.002) at three months.

#### 3.2.2. Lifestyle Habits and Treatment

Regarding lifestyle habits, 77.1% of participants did not use tobacco, alcohol consumption was reported by 8.6% of participants, 97.1% did not report cannabis use, while 2.9% indicated cannabis consumption.

Cannabis use showed interactions with social function, sexual function, and sexual satisfaction at diagnosis; however, since only one individual in the sample reported cannabis use, these results should be interpreted with caution.

Tobacco use was associated with the physical limitation dimension at three months (*p* = 0.045).

Participants received a variety of pharmacological treatments. The most common treatments were Ocrelizumab (20%) and Mavenclad (14.3%). Other treatments included Tecfidera, Tysabri, Vumerity, and Ofatumumab, each accounting for 2.9%, while Alemtuzumab, Cladribine, Corticosteroids, and Diroximel fumarate were administered in 5.7% of cases.

### 3.3. Clinical Results: Quality of Life

#### 3.3.1. Physical Function

The results from Table 2 and Table 3 reveal notable trends in the evolution of QoL across different domains over time. Specifically, physical function demonstrated slight improvements at three months (83.14 ± 13.13) compared to baseline (80.67 ± 17.67), but a subsequent decrease was observed at six months (81.52 ± 12.71). This fluctuation suggests that while initial recovery may occur, sustaining these gains requires targeted interventions to address progressive MS-related challenges.

Homogeneity of variances between diagnosis and 3 months follow-up was significant (F(13,21) = 3.708, *p* = 0.004), indicating that variances were not homogeneous across groups. The corrected model was statistically significant (F(13,21) = 5.776, *p* < 0.001), explaining 78.1% of the variance in the physical function scores (R^2^ = 0.781, adjusted R^2^ = 0.646). This indicates that the independent factor had a considerable impact on the dependent variable. The intercept was also highly significant (F(1,21) = 2627.510, *p* < 0.001), reflecting the importance of the overall mean values of the dependent variable. The analysis revealed a large effect size (*n*^2^ = 0.781), confirming that the differences between groups were substantial. The observed power was excellent (1), indicating sufficient sensitivity to detect significant effects. The grand mean was 79.163 (95% CI: 75.951–82.374), with a standard error of 1.544, providing a reliable estimate of the central tendency.

Levene’s test for equality of variances between physical function at diagnosis and 6 months follow-up was significant (F(13,21) = 5.910, *p* < 0.001), indicating that the assumption of homogeneity of variances was violated. The corrected model for physical function was statistically significant (F(13,21) = 2.685, *p* = 0.021), explaining 62.4% of the variance in the dependent variable (R^2^ = 0.624, adjusted R^2^ = 0.392). This indicates that the independent factor had a moderate impact on the dependent variable. The intercept was highly significant (F(1,21) = 1544.233, *p* < 0.001), reflecting the importance of the overall mean values of the dependent variable. The model demonstrated a moderate effect size (*n*^2^ = 0.624) and high observed power (0.893), suggesting that the analysis had adequate sensitivity to detect significant effects. The grand mean was 77.026 (95% CI: 72.950–81.102), with a standard error of 1.960.

Levene’s test for equality of variances between physical function at 3 and 6 months follow-up was not significant (F(13,21) = 1.677, *p* = 0.144), indicating that the assumption of homogeneity of variances was met. The corrected model was statistically significant (F(13,21) = 10.258, *p* < 0.001), explaining 86.4% of the variance in the dependent variable (R^2^ = 0.864, adjusted R^2^ = 0.780). This indicates that the independent factor had a substantial impact on the dependent variable. The intercept was also highly significant (F(1,22) = 3946.630, *p* < 0.001), reflecting the importance of the overall mean values of the dependent variable. The model demonstrated a large effect size (*n*^2^ = 0.864) and excellent observed power (1), confirming that the analysis had sufficient sensitivity to detect significant effects. The grand mean was 76.833 (95% CI: 74.290–79.377), with a standard error of 1.223.

#### 3.3.2. Physical Limitations

The mean score for physical limitations was 59.57 ± 25.79 at diagnosis, slightly decreased to 57.43 ± 22.17 at three months, and remained stable at 57.43 ± 21.50 at six months.

Levene’s test for equality of variances between physical limitations at diagnosis and 3 months follow-up was significant (F(16,18) = 2.613, *p* = 0.026), indicating that the assumption of homogeneity of variances was met. The corrected model was statistically significant (F(16,18) = 6.644, *p* < 0.001), explaining 85.5% of the variance in the dependent variable (R^2^ = 0.855, adjusted R^2^ = 0.726). This indicates that the independent factor had a substantial impact on the dependent variable. The intercept was also highly significant (F(1,18) = 857.050, *p* < 0.001), reflecting the importance of the overall mean values of the dependent variable. The model demonstrated a large effect size (*n*^2^ = 0.906) and excellent observed power (1), confirming the sensitivity of the analysis to detect significant effects. The grand mean was 58.725 (95% CI: 53.990–66.461), with a standard error of 2.254.

Levene’s test for equality of variances between physical limitations at diagnosis and 6 months follow-up was significant (F(14,20) = 8.473, *p* < 0.001). The corrected model was also statistically significant (F(14,20) = 9.253, *p* < 0.001), explaining 86.6% of the variance in the dependent variable (R^2^ = 0.866, adjusted R^2^ = 0.773). This indicates that the independent factor had a substantial impact on the dependent variable. The intercept was highly significant (F(1,20) = 784.637, *p* < 0.001), and the model demonstrated a large effect size (*n*^2^ = 0.866) and excellent observed power (1). The grand mean was 63.867 (95% CI: 59.111–68.623), with a standard error of 2.280.

Levene’s test for equality of variances between physical limitations at 3 and 6 months follow-up was not significant (F(16,18) = 2.102, *p* = 0.065). The corrected model, however, was statistically significant (F(16,18) = 16.610, *p* < 0.001), explaining 93.7% of the variance in the dependent variable (R^2^ = 0.937, adjusted R^2^ = 0.880). This indicates that the independent factor had a substantial impact on the dependent variable. The intercept was also highly significant (F(1,18) = 1799.499, *p* < 0.001), reflecting the importance of the overall mean values of the dependent variable. The model demonstrated a large effect size (*n*^2^ = 0.937) and excellent observed power (1). The grand mean was 61.456 (95% CI: 58.412–64.500), with a standard error of 1.449.

#### 3.3.3. Emotional and Mental Health Limitations

Emotional limitations showing a reduction from 64.38 ± 23.03 at diagnosis to 58.86 ± 20.48 at six months (effect size d = 0.253, *p* < 0.05). The observed improvements may reflect the benefits of early disease-modifying therapies (DMTs) or psychosocial support; however, the persistence of limitations highlights the need for sustained emotional and psychological interventions.

Levene’s tests for equality of variances between emotional limitations at different time points (diagnosis, 3 months, and 6 months follow-up) yielded non-significant results for all comparisons. Specifically, the test was not significant between diagnosis and 3 months follow-up (F(11,23) = 1.716, *p* = 0.132), between diagnosis and 6 months follow-up (F(11,23) = 1.094, *p* = 0.408), and between 3 and 6 months follow-up (F(12,22) = 1.574, *p* = 0.172), indicating homogeneity of variances across all comparisons.

The corrected model for emotional limitations at diagnosis and 3 months follow-up was statistically significant (F(11,23) = 6.959, *p* < 0.001), explaining 76.9% of the variance in the dependent variable (R^2^ = 0.769, adjusted R^2^ = 0.658). This indicates that the independent factor had a substantial impact on the dependent variable. The intercept was also highly significant (F(1,23) = 399.461, *p* < 0.001), reflecting the importance of the overall mean values of the dependent variable. The model demonstrated a large effect size (*n*^2^ = 0.769) and excellent observed power (1). The grand mean was 57.537 (95% CI: 51.582–63.492), with a standard error of 2.879.

Similarly, the corrected model for emotional limitations at diagnosis and 6 months follow-up was statistically significant (F(11,23) = 4.893, *p* = 0.001), explaining 70.1% of the variance (R^2^ = 0.701, adjusted R^2^ = 0.557). The intercept was also highly significant (F(1,23) = 422.725, *p* < 0.001), and the model demonstrated a large effect size (*n*^2^ = 0.701) and excellent observed power (0.995). The grand mean was 55.788 (95% CI: 50.175–61.401), with a standard error of 2.713.

The corrected model for emotional limitations at 3 and 6 months follow-up was also statistically significant (F(12,22) = 11.627, *p* < 0.001), explaining 86.4% of the variance (R^2^ = 0.864, adjusted R^2^ = 0.790). This indicates that the independent factor had a substantial impact on the dependent variable. The intercept was highly significant (F(1,22) = 962.363, *p* < 0.001), reflecting the importance of the overall mean values of the dependent variable. The model demonstrated a large effect size (*n*^2^ = 0.864) and excellent observed power (1). The grand mean was 56.214 (95% CI: 52.456–59.972), with a standard error of 1.812.

#### 3.3.4. Pain Perception

Pain perception showed slight but clinically relevant improvements at six months (81.79 ± 20.52) compared to baseline (78.93 ± 23.59), emphasizing the importance of pain management in RRMS care. Although these changes are not statistically significant (*p* > 0.05), they underscore the importance of addressing pain management as part of comprehensive care.

Levene’s tests for equality of variances between pain perception at different time points (diagnosis, 3 months, and 6 months follow-up) yielded mixed results. The test was not significant for the comparisons between diagnosis and 3 months follow-up (F(11,23) = 1.280, *p* = 0.296) and between 3 and 6 months follow-up (F(11,23) = 1.280, *p* = 0.296), indicating homogeneity of variances in these cases. However, it was significant for the comparison between diagnosis and 6 months follow-up (F(9,25) = 7.946, *p* < 0.01), suggesting heterogeneity of variances.

The corrected model for pain perception at diagnosis and 3 months follow-up was significant (F(11,23) = 11.049, *p* < 0.001), explaining 84.1% of the variance in the dependent variable (R^2^ = 0.841, adjusted R^2^ = 0.765). This indicates that the independent factor had a substantial impact on the dependent variable. The intercept was also highly significant (F(11,23) = 764.228, *p* < 0.001), reflecting the importance of the overall mean values of the dependent variable. The model demonstrated a large effect size (*n*^2^ = 0.841) and excellent observed power (1). The grand mean was 68.490 (95% CI: 63.364–73.615), with a standard error of 2.477.

Similarly, the corrected model for pain perception at diagnosis and 6 months follow-up was significant (F(9,25) = 10.269, *p* < 0.001), explaining 78.7% of the variance (R^2^ = 0.787, adjusted R^2^ = 0.710). The intercept was also highly significant (F(9,25) = 826.519, *p* < 0.001), and the model demonstrated a large effect size (*n*^2^ = 0.787) and excellent observed power (1). The grand mean was 72.76 (95% CI: 67.548–77.973), with a standard error of 2.531.

The corrected model for pain perception at 3 and 6 months follow-up was also significant (F(11,23) = 11.049, *p* < 0.001), explaining 84.1% of the variance (R^2^ = 0.841, adjusted R^2^ = 0.765). The intercept was highly significant (F(11,23) = 764.228, *p* < 0.001), and the model demonstrated a large effect size (*n*^2^ = 0.841) and excellent observed power (1). The grand mean was 68.490 (95% CI: 63.364–73.615), with a standard error of 2.477.

#### 3.3.5. Mental Health

Mental health scores improved significantly at three months (57.03 ± 17.17) compared to baseline (44.34 ± 14.49), with a moderate effect size (d = −0.799, *p* < 0.05). However, a slight decline was observed at six months (55.09 ± 15.16), suggesting that while initial interventions may alleviate psychological distress, long-term strategies are essential to maintain mental well-being.

Levene’s tests for equality of variances between mental health at different time points (diagnosis, 3 months, and 6 months follow-up) produced mixed results. The test was significant for the comparison between diagnosis and 3 months follow-up (F(14,20) = 4.984, *p* < 0.01), indicating heterogeneity of variances. However, it was not significant for the comparisons between diagnosis and 6 months follow-up (F(14,20) = 1.719, *p* = 0.131) or between 3 and 6 months follow-up (F(13,21) = 1.493, *p* = 0.2), suggesting homogeneity of variances in these cases.

The corrected model for mental health at diagnosis and 3 months follow-up was significant (F(14,20) = 5.973, *p* < 0.001), explaining 80.7% of the variance in the dependent variable (R^2^ = 0.807, adjusted R^2^ = 0.672). This indicates that the independent factor had a substantial impact on the dependent variable. The intercept was highly significant (F(14,20) = 577.984, *p* < 0.001), reflecting the importance of the overall mean values of the dependent variable. The model demonstrated a large effect size (*n*^2^ = 0.807) and excellent observed power (0.999). The grand mean was 61.56 (95% CI: 57.336–65.784), with a standard error of 2.025.

Similarly, the corrected model comparing mental health at diagnosis and 6 months follow-up was significant (F(14,20) = 7.008, *p* < 0.001), explaining 83.1% of the variance (R^2^ = 0.831, adjusted R^2^ = 0.712). The intercept was also highly significant (F(14,20) = 463.434, *p* < 0.001), and the model demonstrated a large effect size (*n*^2^ = 0.831) and excellent observed power (1). The grand mean was 59.244 (95% CI: 55.753–62.736), with a standard error of 1.674.

Finally, the corrected model comparing mental health at 3 and 6 months follow-up was also significant (F(13,21) = 14.206, *p* < 0.001), explaining 89.8% of the variance (R^2^ = 0.898, adjusted R^2^ = 0.835). The intercept was highly significant (F(13,21) = 539.483, *p* < 0.001), reflecting the importance of the overall mean values of the dependent variable. The model demonstrated a large effect size (*n*^2^ = 0.898) and excellent observed power (1). The grand mean was 56.688 (95% CI: 54.1–59.276), with a standard error of 1.244.

#### 3.3.6. Energy

Energy levels showed a mean score of 54.86 ± 20.27 at diagnosis, 55.26 ± 18.63 at three months, and remained stable at six months.

Levene’s tests for equality of variances between energy levels at different time points (diagnosis, 3 months, and 6 months follow-up) were not significant in any of the comparisons: diagnosis and 3 months follow-up (F(14,20) = 1.046, *p* = 0.452), diagnosis and 6 months follow-up (F(14,20) = 1.199, *p* = 0.347), and 3 and 6 months follow-up (F(13,21) = 1.488, *p* = 0.202). These results indicate homogeneity of variances across all comparisons.

The corrected model for energy levels at diagnosis and 3 months follow-up was significant (F(14,20) = 14.377, *p* < 0.001), explaining 91% of the variance in the dependent variable (R^2^ = 0.91, adjusted R^2^ = 0.846). This indicates that the independent factor had a substantial impact on the dependent variable. The intercept was highly significant (F(14,20) = 766.781, *p* < 0.001), reflecting the importance of the overall mean values of the dependent variable. The model demonstrated a large effect size (*n*^2^ = 0.910) and excellent observed power (1). The grand mean was 55.778 (95% CI: 52.861–58.695), with a standard error of 1.398.

Similarly, the corrected model comparing energy levels at diagnosis and 6 months follow-up was significant (F(14,20) = 6.742, *p* < 0.001), explaining 82.5% of the variance (R^2^ = 0.825, adjusted R^2^ = 0.703). The intercept was also highly significant (F(14,20) = 506.971, *p* < 0.001), and the model demonstrated a large effect size (*n*^2^ = 0.825) and excellent observed power (1). The grand mean was 55.022 (95% CI: 51.558–58.486), with a standard error of 1.661.

Finally, the corrected model comparing energy levels at 3 and 6 months follow-up was also significant (F(13,21) = 14.246, *p* < 0.001), explaining 89.8% of the variance (R^2^ = 0.898, adjusted R^2^ = 0.835). The intercept was highly significant (F(13,21) = 594.277, *p* < 0.001), reflecting the importance of the overall mean values of the dependent variable. The model demonstrated a large effect size (*n*^2^ = 0.898) and excellent observed power (1). The grand mean was 56.5 (95% CI: 53.867–59.133), with a standard error of 1.266.

#### 3.3.7. Health Perception

Health perception had a mean score of 48.11 ± 18.99 at diagnosis, 47.54 ± 16.77 at three months, and showed a slight improvement to 48.91 ± 17.16 at six months.

Levene’s tests for equality of variances between health perception at different time points (diagnosis, 3 months, and 6 months follow-up) produced mixed results. The test was significant for the comparison between diagnosis and 3 months follow–up (F(14,20) = 4.434, *p* < 0.01), indicating heterogeneity of variances. However, it was not significant for the comparisons between diagnosis and 6 months follow-up (F(14,20) = 1.357, *p* = 0.26) or between 3 and 6 months follow-up (F(13,21) = 1.694, *p* = 0.137), suggesting homogeneity of variances in these cases.

The corrected model for health perception at diagnosis and 3 months follow-up was significant (F(14,20) = 10.138, *p* < 0.001), explaining 87.6% of the variance in the dependent variable (R^2^ = 0.876, adjusted R^2^ = 0.79). This indicates that the independent factor had a substantial impact on the dependent variable. The intercept was highly significant (F(14,20) = 598.563, *p* < 0.001), reflecting the importance of the overall mean values of the dependent variable. The model demonstrated a large effect size (*n*^2^ = 0.876) and excellent observed power (1). The grand mean was 49.209 (95% CI: 46.134–52.284), with a standard error of 1.474.

Similarly, the corrected model comparing health perception at diagnosis and 6 months follow-up was significant (F(14,20) = 7.134, *p* < 0.001), explaining 83.3% of the variance (R^2^ = 0.833, adjusted R^2^ = 0.716). The intercept was highly significant (F(14,20) = 596.234, *p* < 0.001), and the model demonstrated a large effect size (*n*^2^ = 0.833) and excellent observed power (1). The grand mean was 50.809 (95% CI: 47.15–54.468), with a standard error of 1.754.

Finally, the corrected model comparing health perception at 3 and 6 months follow-up was also significant (F(13,21) = 10.834, *p* < 0.001), explaining 87% of the variance (R^2^ = 0.87, adjusted R^2^ = 0.79). The intercept was highly significant (F(13,21) = 670.673, *p* < 0.001), reflecting the importance of the overall mean values of the dependent variable. The model demonstrated a large effect size (*n*^2^ = 0.87) and excellent observed power (1). The grand mean was 50.214 (95% CI: 47.273–53.156), with a standard error of 1.414.

#### 3.3.8. Social Function

Social function had a mean score of 70.48 ± 16.67 at diagnosis, while cognitive function scored 70.12 ± 21.40. At three months, these variables slightly decreased, reaching mean scores of 66.86 ± 15.55 and 69.04 ± 19.78, respectively. At six months, social function worsened further compared to previous periods, reaching a mean score of 65.52 ± 12.78.

Levene’s tests for equality of variances between social function at different time points (diagnosis, 3 months, and 6 months follow-up) produced mixed results. The test was not significant for comparisons between diagnosis and 3 months follow-up (F(9,25) = 1.056, *p* = 0.427) and between diagnosis and 6 months follow-up (F(9,25) = 1.822, *p* = 0.114), indicating homogeneity of variances in these cases. However, it was significant for the comparison between 3 and 6 months follow-up (F(8,26) = 3.440, *p* < 0.01), suggesting heterogeneity of variances.

The corrected models for comparisons involving social function at diagnosis and follow-up time points at 3 and 6 months were not significant (F(9,25) = 1.571, *p* = 0.158 and F(9,25) = 1.092, *p* = 0.403, respectively), explaining 36.1% (R^2^ = 0.361, adjusted R^2^ = 0.131) and 28.2% (R^2^ = 0.282, adjusted R^2^ = 0.024) of the variance in the dependent variable. The intercepts for both models were significant (F(9,25) = 444.292, *p* < 0.001 and F(9,25) = 574.314, *p* < 0.001, respectively), reflecting the importance of the overall mean values of the dependent variable. The grand means were 67.229 (95% CI: 60.66–73.797, SE = 3.189) and 66.203 (95% CI: 60.879–72.327, SE = 2.779), respectively. Both models demonstrated moderate observed power (0.584 and 0.412) and large effect sizes (*n*^2^ = 0.361 and *n*^2^ = 0.282).

Conversely, the corrected model comparing social function at 3 and 6 months follow-up was significant (F(8,26) = 3.448, *p* < 0.01), explaining 51.5% of the variance (R^2^ = 0.515, adjusted R^2^ = 0.365). The grand mean was 65.905 (95% CI: 61.875–69.936), with a standard error of 1.961. The intercept was highly significant (F(8,26) = 1129.889, *p* < 0.001), and the model demonstrated a large effect size (*n*^2^ = 0.515) and strong observed power (0.927).

#### 3.3.9. Cognitive Function

Cognitive function at six months declined to a mean score of 67.50 ± 19.30. Levene’s tests for equality of variances between cognitive function at different time points (diagnosis, 3 months, and 6 months follow-up) were significant in all comparisons, indicating heterogeneity of variances. Specifically, the test was significant for comparisons between diagnosis and 3 months follow-up (F(12,22) = 3.065, *p* = 0.011), between diagnosis and 6 months follow-up (F(12,22) = 2.474, *p* = 0.032), and between 3 and 6 months follow-up (F(13,21) = 7.042, *p* < 0.01).

The corrected model comparing cognitive function at diagnosis and 3 months follow-up was significant (R^2^ = 0.856, adjusted R^2^ = 0.777). The grand mean was 67.81 (95% CI: 64.281–71.338), with a standard error of 1.701. The intercept was highly significant (F(12,22) = 1588.549, *p* < 0.001), and the model demonstrated a large effect size (*n*^2^ = 0.856) and maximum observed power (1).

Similarly, the corrected model comparing cognitive function at diagnosis and 6 months follow-up was significant (F(12,22) = 9.153, *p* < 0.01), explaining 83.3% of the variance (R^2^ = 0.833, adjusted R^2^ = 0.742). The grand mean was 63.266 (95% CI: 62.560–69.972), with a standard error of 1.787. The intercept was highly significant (F(12,22) = 1375.201, *p* < 0.001), and the model demonstrated a large effect size (*n*^2^ = 0.833) and maximum observed power (1).

Finally, the corrected model comparing cognitive function at 3 and 6 months follow-up was also significant (F(13,21) = 14.075, *p* < 0.01), explaining 89.7% of the variance (R^2^ = 0.897, adjusted R^2^ = 0.833). The grand mean was 63.943 (95% CI: 60.678–67.209), with a standard error of 1.57. The intercept was highly significant (F(13,21) = 1658.014, *p* < 0.001), and the model demonstrated a large effect size (*n*^2^ = 0.897) and maximum observed power (1).

#### 3.3.10. Stress

Stress had a mean score of 44.29 ± 22.51 at diagnosis, which increased to 47.38 ± 18.40 at three months and further to 48.95 ± 17.84 at six months, showing a trend toward improvement.

Levene’s tests for equality of variances between stress levels at different time points (diagnosis, 3 months, and 6 months follow-up) produced mixed results. The test was not significant for comparisons between diagnosis and 3 months follow-up (F(14,20) = 1.305, *p* = 0.286) and between 3 and 6 months follow-up (F(14,20) = 0.931, *p* = 0.546), indicating homogeneity of variances. However, it was significant for the comparison between diagnosis and 6 months follow-up (F(14,20) = 3.286, *p* < 0.01), suggesting heterogeneity of variances.

The corrected model comparing stress levels at diagnosis and 3 months follow-up was significant (F(14,20) = 3.380, *p* < 0.01), explaining 70.3% of the variance in the dependent variable (R^2^ = 0.703, adjusted R^2^ = 0.495). The grand mean was 53.241 (95% CI: 47.831–58.65), with a standard error of 2.593. The intercept was highly significant (F(14,20) = 421.501, *p* < 0.001), and the model demonstrated a large effect size (*n*^2^ = 0.703) and strong observed power (0.959).

Similarly, the corrected model comparing stress levels at diagnosis and 6 months follow-up was significant (F(14,20) = 3.470, *p* < 0.01), explaining 70.8% of the variance (R^2^ = 0.708, adjusted R^2^ = 0.504). The grand mean was 52.407 (95% CI: 47.217–57.598), with a standard error of 2.488. The intercept was highly significant (F(14,20) = 443.630, *p* < 0.001), and the model demonstrated a large effect size (*n*^2^ = 0.708) and strong observed power (0.964).

For the comparison between stress levels at 3 and 6 months follow-up, although Levene’s test was not significant (*p* = 0.546), the corrected model was significant (F(14,20) = 5.236, *p* < 0.01), explaining 78.6% of the variance (R^2^ = 0.786, adjusted R^2^ = 0.636). The grand mean was 46 (95% CI: 41.584–50.416), with a standard error of 2.117. The intercept was highly significant (F(14,20) = 443.630, *p* < 0.001), and the model demonstrated a large effect size (*n*^2^ = 0.786) and maximum observed power (0.998).

#### 3.3.11. Sexual Function

Sexual function showed a slight decline, dropping from 79.46 ± 17.12 at diagnosis to 78.21 ± 18.96 at three months and 78.93 ± 18.25 at six months.

Levene’s tests for equality of variances between sexual function at different time points (diagnosis, 3 months, and 6 months follow-up) yielded mixed results. The test was significant for comparisons between diagnosis and 3 months follow-up (F(8,26) = 10.246, *p* < 0.01) and between diagnosis and 6 months follow-up (F(8,26) = 3.898, *p* < 0.01), indicating heterogeneity of variances. However, the test was not significant for the comparison between 3 and 6 months follow-up (F(9,25) = 3.013, *p* = 0.546).

The corrected model comparing sexual function at diagnosis and 3 months follow-up was significant (F(8,26) = 13.580, *p* < 0.01), explaining 88.8% of the variance in the dependent variable (R^2^ = 0.878, adjusted R^2^ = 0.841). The grand mean was 72.234 (95% CI: 69.389–75.079), with a standard error of 1.384. The intercept was highly significant (F(8,26) = 2723.776, *p* < 0.001), and the model demonstrated a large effect size (*n*^2^ = 0.878) and maximum observed power (1).

Similarly, the corrected model comparing sexual function at diagnosis and 6 months follow-up was significant (F(8,26) = 13.580, *p* < 0.01), explaining 80.7% of the variance (R^2^ = 0.807, adjusted R^2^ = 0.747). The grand mean was 74.352 (95% CI: 70.902–77.801), with a standard error of 1.678. The intercept was highly significant (F(8,26) = 1963.1, *p* < 0.001), and the model demonstrated a large effect size (*n*^2^ = 0.807) and maximum observed power (1).

For the comparison between sexual function at 3 and 6 months follow-up, although Levene’s test was not significant (*p* = 0.546), the corrected model was significant (F(9,25) = 18.909, *p* < 0.01), explaining 87.2% of the variance (R^2^ = 0.872, adjusted R^2^ = 0.826). The grand mean was 71.292 (95% CI: 67.953–74.63), with a standard error of 1.621. The intercept was highly significant (F(9,25) = 1934.138, *p* < 0.001), and the model demonstrated a large effect size (*n*^2^ = 0.872) and maximum observed power (1).

#### 3.3.12. Health Changes Dimension

Health changes, as measured at three and six months, showed significant improvements (d = −0.566, *p* < 0.05). These findings align with the literature emphasizing the cumulative benefits of early interventions in mitigating disease progression and improving patient-reported outcomes.

Levene’s tests for equality of variances between health changes at different time points (diagnosis, 3 months, and 6 months follow-up) were not significant, indicating no substantial differences in variance across these intervals.

The corrected model comparing health changes at diagnosis and 3 months follow-up was significant (F(3,31) = 4.626, *p* < 0.01), explaining 30.9% of the variance (R^2^ = 0.309, adjusted R^2^ = 0.242). The grand mean was 48.022 (95% CI: 38.388–57.656), with a standard error of 4.723. The intercept was highly significant (F(3,31) = 103.361, *p* < 0.001), and the model demonstrated a large effect size (*n*^2^ = 0.309) and strong observed power (0.848).

In contrast, the models comparing health changes at diagnosis and 6 months follow-up, as well as between 3 and 6 months follow-up, were not significant and are not further reported, as they do not contribute relevant findings.

#### 3.3.13. Sexual Satisfaction

Sexual satisfaction was scored at 65.14 ± 23.93 at diagnosis, 65.71 ± 20.33 at three months, and slightly worsened to 64.15 ± 20.20 at six months.

Levene’s tests for equality of variances between sexual satisfaction at different time points (diagnosis, 3 months, and 6 months follow-up) were not significant, indicating no substantial differences in variance across these intervals. Despite this, the corrected models for all comparisons were significant, explaining a considerable proportion of the variance in the dependent variable. Specifically, the model between sexual satisfaction at diagnosis and 3 months follow-up was significant (F(4,30) = 9.624, *p* < 0.01), explaining 56.2% of the variance (R^2^ = 0.562, adjusted R^2^ = 0.504) with a grand mean of 64.162 (95% CI: 58.743–69.58) and a standard error of 2.653. Similarly, the model between diagnosis and 6 months follow-up yielded identical results (F(4,30) = 9.624, *p* < 0.01; R^2^ = 0.562, adjusted R^2^ = 0.504) with the same grand mean and standard error values.

In contrast, the model comparing sexual satisfaction at 3 and 6 months follow-up explained a larger proportion of the variance (R^2^ = 0.697, adjusted R^2^ = 0.656) and demonstrated a stronger effect size (*n*^2^ = 0.697) with an observed power of 1. This model was also significant (F(4,30) = 17.212, *p* < 0.01), with a grand mean of 63.719 (95% CI: 57.661–69.776) and a standard error of 2.966. Across all models, the intercepts were highly significant (*p* < 0.001), reflecting the importance of the overall mean values of the dependent variable.

#### 3.3.14. Perceived Quality of Life

While most HRQoL dimensions remained stable, domains such as emotional limitations and social functioning exhibited progressive decline, suggesting cumulative psychological and social burdens over time. HRQoL at diagnosis showed a mean score of 62.39 ± 14.15 points, slightly increasing to 62.84 ± 12.72 points at three months and remaining stable at six months with a mean score of 62.40 ± 11.80 points (Figure 2).

The corrected Levene’s models assessing HRQoL at different time points showed varying levels of significance and variance explained. The model between HRQoL at diagnosis and at 3 months was not significant (F(31,3) = 7.318, *p* = 0.063), explaining 98.7% of the variance in the dependent variable (R^2^ = 0.987, adjusted R^2^ = 0.852), with a grand mean of 163.516 (95% CI: 156.478–170.553) and a standard error of 2.211. In contrast, the model between HRQoL at diagnosis and at 6 months was significant (F(31,3) = 11.372, *p* < 0.01), explaining 99.2% of the variance (R^2^ = 0.992, adjusted R^2^ = 0.904). The intercept was highly significant (F(31,3) = 9798.676, *p* < 0.001), with a large effect size (*n*^2^ = 0.992) and adequate observed power (0.767). The grand mean for this model was 162.969 (95% CI: 157.729–168.208), with a standard error of 1.646. Finally, the model comparing HRQoL at 3 and 6 months was not significant (F(32,2) = 1.910, *p* = 0.403), indicating no substantial differences in variance. This model explained 96.8% of the variance (R^2^ = 0.968, adjusted R^2^ = 0.461), with a grand mean of 163.939 (95% CI: 147.242–180.637) and a standard error of 3.881.

Visual representations of the data (Figure 3) illustrate the trends across time points for the QoL dimensions. Line graphs highlight subtle changes in dimensions such as mental health, stress, and health changes, which showed slight improvements over time. Confidence intervals for these trends have been added to enhance interpretability.

The paired sample analysis revealed no statistically significant differences in QoL scores across the three time points measured (baseline, 3 months, and 6 months). Effect sizes were minimal across all comparisons, ranging from 0.00 to 0.10, indicating negligible clinical relevance of the observed differences. In contrast, the paired sample correlations demonstrated very strong and statistically significant relationships between the QoL scores at the three time points. These findings suggest a high degree of consistency in QoL measurements over time, despite the lack of significant changes between the time points. (Table 4 and Table 5).

In summary, this study evaluated the sociodemographic and clinical characteristics, as well as the QoL, of a sample of 35 individuals with RRMS across three time points: baseline (diagnosis), three months, and six months. The specific QoL dimensions analyzed included physical function, physical limitations, emotional limitations, pain perception, mental health, energy, health perception, social function, cognitive function, stress, sexual function, health changes, satisfaction with sexual life, and HRQoL.

Although no statistically significant differences were observed in overall HRQoL across the three time points, significant differences were identified in several specific QoL dimensions. For instance, age was significantly associated with QoL at baseline (*p* = 0.018) and at three months (*p* = 0.024), while employment status showed significant correlations with multiple dimensions, including energy (*p* = 0.024 at baseline and *p* = 0.001 at six months) and social function (*p* = 0.012 at three months and *p* = 0.001 at six months). Additionally, χ^2^ analyses revealed significant differences between age groups in terms of QoL at baseline (χ ^2^ = 8.789; *p* = 0.012) and at three months (χ^2^ = 8.415; *p* = 0.015).

Temporal trends showed slight improvements in specific dimensions, such as health changes and stress, while others, such as emotional limitations and social function, exhibited minor declines. Effect sizes, measured using Cramer’sV (V = 0.24 and V = 0.25), indicated moderate associations.

Finally, correlation analyses demonstrated strong consistency in QoL measurements over time, with Spearman’s correlation coefficients being high and statistically significant (*p* < 0.01). These findings emphasize the importance of longitudinal monitoring of QoL in this population, particularly in dimensions showing significant variations. Furthermore, the results highlight the need to investigate the sociodemographic and clinical factors influencing QoL trajectories in individuals with RRMS.

## 4. Discussion

### 4.1. Relationships Between Sociodemographic Variables and Quality of Life

The results revealed a significant relationship between age and QoL at diagnosis and at three months. However, this relationship was not sustained at six months. Younger participants might perceive a better initial QoL, possibly due to fewer comorbidities and greater physical and psychological recovery capacity in the early stages of the disease. Gil-González et al. [53] identified age as a relevant factor in the HRQoL, especially during the initial phases of the disease, where younger individuals tend to report better physical and emotional well-being. Moreover, the cumulative burden of the disease in older individuals negatively impacts their perception of well-being [54]. This finding underscores the importance of early interventions tailored to address the specific needs of older individuals, who may face greater challenges in maintaining QoL due to disease burden.

This observation is supported by multiple studies. Tarasiuk et al. [55] reported that younger age was significantly associated with higher QoL scores, particularly in the EQ-5D-5L index (mean score: 0.89 ± 0.15) and the EQ-Visual Analog Scale scores (VAS) (mean score: 71.58 ± 18.67). These findings align with our conclusion that younger individuals experience better HRQoL, likely due to fewer comorbidities and lower disability levels. Similarly, Castillo-Zuñiga et al. [56] corroborated this relationship, reporting a mean age of 36.5 ± 8.9 years among their RRMS cohort, with younger individuals demonstrating better cognitive performance and QoL.

Song et al. [57] also highlighted the positive relationship between younger age and HRQoL in Chinese MS patients, noting higher EQ-5D-5L health utility scores (Health Utility Scores; mean: 0.73, SD: 0.29) and EQ-VAS (mean: 68.7, SD: 23.9) in younger patients. These results further emphasize that younger individuals tend to experience fewer comorbidities and better recovery capacity in the early stages of MS. Additionally, Nauta et al. [58] reported an average participant age of 46.2 ± 10.5 years, with younger individuals showing greater compliance with a multi-domain lifestyle intervention, which was associated with improved QoL outcomes. Their findings suggest that younger patients may benefit more from lifestyle changes, as they often demonstrate better recovery capacity and adaptability, which positively influence their QoL. However, the specific mechanisms underlying these age-related differences warrant further investigation, particularly in longitudinal studies that explore the interplay between age, treatment response, and lifestyle factors.

Although women represented 57.1% of the sample, no significant differences were observed between men and women in relation to QoL. Research indicates that, despite the higher prevalence of MS among women, differences in QoL between sexes are more influenced by factors such as social support and symptom severity rather than sex itself [54,59]. This aligns with the study by Sabanagic-Hajric et al. [60], which highlights that gender does not have a direct impact on QoL but does affect specific domains, such as sexual function, influenced by clinical and social factors.

This trend was also observed in other studies. Castillo-Zuñiga et al. [56] reported a higher proportion of females in their sample (73.7%), reflecting the known female predominance in MS. However, their study found no significant differences in cognitive performance or QoL between sexes, reinforcing the idea that gender alone is not a determinant of QoL in MS. Similarly, Tarasiuk et al. [55] reported a female-to-male ratio of 2.2:1 in their study population, but gender differences did not significantly impact overall QoL scores.

Song et al. [57] also observed a higher prevalence of MS among females (67.7%), consistent with global epidemiological trends, yet their findings did not identify significant differences in HRQoL between genders. Nauta et al. [58] reported a female predominance in their cohort (84.5% women) and similarly found no significant gender-based differences in QoL outcomes. These consistent findings across studies reinforce the conclusion that sex alone does not significantly influence overall QoL in MS patients. However, the potential influence of gender-specific factors, such as hormonal changes or reproductive concerns, on specific QoL dimensions (e.g., emotional and sexual function) should not be overlooked.

Married participants showed a significant interaction with stress at diagnosis. Marital status could influence stress perception due to the availability of emotional support, as suggested by previous research. Uhr et al. [61] identified marital status and social support as protective factors against anxiety and depression in individuals with MS, directly influencing their HRQoL. However, the variability in marital status effects across studies highlights the need for more nuanced analyses that account for the quality and dynamics of interpersonal relationships.

Similarly, educational level showed a correlation with cognitive function at diagnosis and at three months, supporting studies indicating that higher educational attainment may be associated with better cognitive strategies for coping with the disease [59,60]. Reece et al. [62] noted that higher education levels facilitate the adoption of health education programs and coping strategies, thereby improving HRQoL.

This correlation is further supported by Castillo-Zuñiga et al. [56], whose inclusion criterion of at least one year of formal education underscores the importance of education in cognitive performance and QoL. Their findings demonstrated significant correlations between verbal and visuospatial memory (*r* = 0.668, *p* = 0.002), suggesting that cognitive domains are interdependent and benefit from higher educational attainment. Song et al. [57] also emphasized the protective role of higher education levels in HRQoL, identifying it as a factor significantly associated with better Health Utility and pain scores. Specifically, patients with higher education levels demonstrated better coping mechanisms and reported higher QoL scores, consistent with our findings. These findings highlight the potential of educational interventions, such as tailored health literacy programs, to enhance cognitive and emotional resilience in MS patients.

In contrast, Nauta et al. [58] observed that participants with lower education levels experienced greater improvements in mental functioning following the lifestyle intervention (β = −3.48, *p* < 0.001). This suggests that targeted interventions may be particularly beneficial for individuals with lower educational attainment, facilitating improvements in coping strategies and overall QoL. Further research is needed to explore how educational attainment interacts with other sociodemographic and clinical factors to shape QoL trajectories over time.

The sample size calculation (35 patients) is based on the population residing in the Lugo health area. Therefore, the results are representative of this specific population, which should be considered when interpreting the findings. This limitation highlights the need for caution when generalizing the results to broader populations and should be addressed in future research.

### 4.2. Impact of Clinical Characteristics

A significant relationship was observed between family history and mental health at diagnosis, as well as with health changes at three months. This may reflect the psychological impact of having a family history of severe illnesses, as documented in studies analyzing the emotional effects of chronic diseases within family environments [54,59]. Uhr et al. [61] emphasized that family stress associated with chronic diseases can negatively influence mental health perception. This highlights the importance of addressing familial dynamics and stressors in the comprehensive care of MS patients.

The psychological impact of family history was further highlighted by Lindberg et al. [63], who demonstrated that progression independent of relapse activity (PIRA) individuals had significantly worse mental health outcomes, as measured by the 29-item Multiple Sclerosis Impact Scale, compared to non-PIRA individuals (*p* = 0.036). This reinforces the need to consider family history and psychological factors when addressing mental health in RRMS patients. Moreover, incorporating family-based interventions, such as psychoeducational sessions and support groups, could mitigate the emotional burden associated with familial stressors.

The findings underscore the importance of considering the family context in the comprehensive care of these individuals, as stress associated with chronic diseases within the family unit can negatively affect mental health perception [53,64]. It would be pertinent to design specific interventions aimed at providing psychological support to both patients and their families, with the goal of mitigating emotional impact and improving HRQoL. For instance, implementing family support groups or psychoeducation sessions could help reduce associated stress and foster more effective coping strategies [53,61,64].

Autoimmune diseases showed significant interaction with physical function at diagnosis, as well as with stress and health changes at three months, and were further associated with pain at six months. Recent research has indicated that autoimmune comorbidities can exacerbate symptoms and negatively affect QoL, particularly in physical and psychological dimensions [60]. Gil-González et al. [53] emphasized that autoimmune comorbidities have a cumulative impact on QoL, especially in physical and emotional domains.

Song et al. [57] identified disability status, as measured by the Expanded Disability Status Scale (EDSS), as a major determinant of HRQoL in MS patients. Their findings showed that worse disability status was significantly associated with lower Health Utility and pain scores (*p* < 0.001). This aligns with our observation that clinical factors such as comorbidities and disease progression play a critical role in shaping patients’ QoL. Additionally, anxiety and depression were highly prevalent (reported by 74.9% of patients), further emphasizing the psychological burden of MS.

Nauta et al. [58] demonstrated that participants with obesity experienced the greatest improvements in physical functioning following the lifestyle intervention (β = −2.50, *p* < 0.001), highlighting the importance of addressing comorbidities as part of comprehensive MS care. This finding aligns with our observation that clinical factors, such as disease progression and comorbidities, play a critical role in shaping QoL outcomes.

Prior mononucleosis was associated with social function at diagnosis and sexual function. This aligns studies suggesting that previous infections, such as mononucleosis, may have residual impacts on social and sexual health in individuals with MS [59]. Sabanagic-Hajric et al. [60] highlighted that clinical factors, such as prior infections, can influence sexual and social function in individuals with MS.

Castillo-Zuñiga et al. [56] further supported the impact of clinical characteristics on QoL, reporting significant differences in total QoL across treatment groups (F = 8.007, *p* = 0.029). Additionally, their findings demonstrated strong correlations between overall QoL and general health perception (r = 0.793, *p* < 0.001), emphasizing the interconnectedness of clinical and psychosocial factors.

Tarasiuk et al. [55] identified significant associations between lower disability levels (EDSS ≤ 3.5), shorter disease duration, and higher QoL scores (*p* < 0.001). This finding is consistent with our observation that clinical factors, such as comorbidities and disease progression, play a crucial role in shaping patients’ QoL. Notably, their study also highlighted that pain/discomfort (80.7%) and anxiety/depression (79.6%) were the most frequently reported problems, further emphasizing the multidimensional nature of MS-related challenges.

Although cannabis use showed interaction with several dimensions, the low frequency of consumption within the sample limits the interpretation of these results. On the other hand, tobacco use was associated with physical limitations at three months, highlighting its negative impact on physical health, as also noted in studies on lifestyle habits and QoL in individuals with neurological diseases [54]. Reece et al. [62] identified that habits such as tobacco use negatively affect the perception of QoL, particularly in physical dimensions.

Lifestyle habits, such as tobacco, alcohol, and cannabis use, have a significant impact on various dimensions of QoL. In the case of tobacco, the results show that its consumption is associated with physical limitations at three months, possibly reflecting its negative effects on respiratory and cardiovascular capacity, exacerbating fatigue and reducing energy available for daily activities. On the other hand, although alcohol consumption was low in the sample, previous research has indicated that even moderate consumption can negatively influence cognitive and social function, affecting HRQoL. Cannabis use, though limited to a single participant in this study, showed interaction with social function, sexual function, and sexual satisfaction, which may be related to its effects on mood and perceived well-being [62,65,66].

From a preventive approach, it would be useful to implement educational programs aimed at promoting healthy lifestyle habits in individuals with RRMS, emphasizing the risks associated with substance use and its impact on QoL. Additionally, interventions focused on promoting physical exercise, a balanced diet, and stress management techniques could counteract the negative effects of these habits and improve physical, emotional, and social dimensions of QoL [62,65,66].

### 4.3. Lifestyle Habits and Quality of Life

A significant relationship was found between pregnancy planning and sexual satisfaction and function at three months, which may reflect greater concern for reproductive aspects among individuals considering pregnancy. This has also been reported in studies on QoL and reproductive health in women with MS [59]. Sabanagic-Hajric et al. [60] confirmed that reproductive planning has a significant impact on sexual function and the perception of QoL, highlighting the importance of addressing these concerns in the comprehensive care of women with MS. However, it is essential to explore whether reproductive planning interacts with other dimensions of QoL, such as emotional well-being or stress, particularly in populations with diverse cultural or socioeconomic backgrounds.

Nauta et al. [58] further emphasized the significant impact of a multi-domain lifestyle intervention on physical activity, diet, stress, and sleep management, reporting improvements in both physical and mental functioning. Specifically, participants experienced reductions in MS-related symptoms and enhanced overall QoL (β = −2.44, *p* < 0.001). This finding underscores the importance of targeting multiple lifestyle factors simultaneously to achieve optimal outcomes in MS management, particularly for addressing the multidimensional challenges faced by MS patients. The integration of such interventions into routine care, especially through personalized strategies, could amplify their effectiveness and long-term adherence.

Energy levels showed a significant correlation with employment status at diagnosis, three months, and six months. Social function and stress also interacted with employment status at different time points, suggesting that the type of work may influence overall HRQoL. Gil-González et al. [53] highlighted that job stability acts as a protective factor for QoL by providing structure and social support. This aligns with studies indicating that employment can serve as a protective factor in quality-of-life perception, offering both structure and social support [60]. Nevertheless, further research is needed to disentangle the specific aspects of employment—such as job satisfaction, work flexibility, or financial stability—that most strongly influence QoL in MS patients.

### 4.4. Evolution of Quality of Life

The impact of MS on QoL has been extensively documented in the literature. The study by Hernández et al. [37], focused exclusively on quality-of-life assessment, observed how this is influenced by the type of MS, the degree of disability, and the time elapsed since diagnosis. Similarly, Castillo-Zuñiga et al. [56] reported that fatigue with a median score of 60 points (IQR 31.75–73.25), indicating a high prevalence of fatigue in their cohort, which aligns with previous findings on its disabling impact on QoL. Fatigue, as one of the most pervasive symptoms of MS, has been consistently linked to diminished physical functioning and emotional well-being, underscoring the need for targeted interventions to manage this symptom.

Tarasiuk et al. [55] observed that the overall QoL in Polish MS patients, as measured by the EQ-5D-5L and EQ-VAS, has improved over the past decade. This improvement is attributed to advances in MS care, including broader access to DMTs and optimized treatment strategies. Their findings highlight that frequent DMT switches are associated with lower QoL (*p* < 0.001), suggesting the importance of stable and effective treatment plans. This aligns with the present study’s observation of stable QoL scores over time, which may reflect the role of consistent therapeutic approaches in maintaining overall well-being.

In addition, Song et al. [57] reported that MS subtypes significantly influenced HRQoL, with secondary progressive MS (SPMS) patients reporting worse VAS compared to RRMS patients. This finding underscores the importance of early intervention and tailored management strategies to mitigate the progressive decline in QoL associated with advanced MS subtypes. The differentiation of care strategies based on MS subtype is critical, as it allows for a more precise allocation of resources and interventions.

Nauta et al. [58] observed that the impact of MS on daily functioning reduced significantly following the lifestyle intervention, with sustained improvements observed at the 3-month follow-up (β = −2.00, *p* < 0.001). This finding highlights the potential for lifestyle interventions to mitigate the progressive decline in QoL associated with MS, particularly when implemented over sustained periods. Such interventions could be further enhanced by incorporating technological tools, such as mobile applications, to provide continuous support and monitoring.

The mean HRQoL score of approximately 62 points observed in this study reflects a moderate level of quality of life, which aligns with findings reported in similar cohorts [51,56]. This score indicates that, while patients maintain a degree of functionality and well-being, significant challenges persist, particularly in domains such as emotional and social functioning. Previous studies have highlighted that scores in this range are often associated with substantial limitations in daily activities, emotional distress, and reduced social participation [55,57]. Such challenges may stem from the cumulative burden of physical symptoms, fatigue, and psychological factors, which are characteristic of MS and tend to worsen over time in the absence of targeted interventions. These findings reinforce the need for holistic care models that address both the physical and psychosocial dimensions of MS.

HRQoL remained relatively stable during the period analyzed, with an average score of approximately 62 points. However, significant positive correlations between QoL at diagnosis, three months, and six months suggest that individuals with better initial QoL tend to maintain this perception over time. Previous studies have reported similar patterns, where initial QoL strongly predicts future evolution [54,60]. This stability highlights the importance of identifying predictors of high initial QoL, as these factors may serve as targets for early interventions.

The observed decline in emotional and social domains further underscores the multifaceted impact of MS on patients’ lives. Emotional limitations, often exacerbated by anxiety, depression, and uncertainty about disease progression, contribute to a diminished sense of well-being [59]. Similarly, reduced social functioning may reflect the challenges patients face in maintaining relationships and participating in social activities due to fatigue, physical limitations, or stigma [53,55]. These findings highlight the need for comprehensive care models that address not only physical symptoms but also the psychological and social dimensions of MS. Integrating psychosocial support into routine care could mitigate the negative trajectory observed in these domains.

The evolution of results in the present study aligns with those described by Martínez-Espejo et al. [51], who, after applying the MSQOL-54 to a sample of MS patients, reported a global mean score of 61.1 points (SD = 20.06), with a wide range of scores between 13.9 and 97.2. These data not only indicate significant inter-individual variability but also reflect a generalized and significant impact on QoL from early stages. The similarity between the results of Martínez-Espejo et al. [51] and those observed in this study reinforces the reliability of the identified deterioration pattern and provides external consistency to the phenomenon described herein.

Additionally, Kobelt et al. [66] conducted an analysis of QoL and associated costs of MS in Spain, revealing that QoL decreases significantly as disability progresses. This underscores the importance of early interventions and continuous care to mitigate cumulative deterioration.

Lindberg et al. [63] reported a significant deterioration in QoL over a five-year period in PIRA individuals, as evidenced by lower EQ-5D-3L scores (*p* = 0.001) and higher MSIS-29 PHYS scores (*p* = 0.004). Additionally, delta values indicated a significant decline in EQ-VAS (*p* = 0.010) and MSIS-29 PSYCH (*p* = 0.036) scores in PIRA individuals compared to non-PIRA individuals.

Although some dimensions, such as physical function and pain perception, showed slight improvement over time, others, such as emotional limitations and social function, worsened progressively. This may reflect the cumulative psychological and social burden of the disease, as also reported in longitudinal studies on QoL in MS patients [59]. While the mentioned studies consistently highlight moderate to severe impairment of QoL from early stages, it is also important to analyze research where interventions targeting physical and emotional distress have been implemented.

In this regard, a study conducted by Fidao et al. [67] reinforces the need for early intervention by demonstrating that the adoption of several healthy lifestyle habits is associated with improvements in both physical and mental QoL in MS patients. This comparison also suggests that improving QoL does not necessarily require complex or costly interventions but can be promoted through simple and sustained recommendations over time.

Moreover, a literature review by Montañés-Masias et al. [68] provides a complementary perspective, focusing on psychological approaches through the evaluation of the effectiveness of online interventions targeting MS patients. Their findings concluded that programs such as mindfulness, cognitive-behavioral therapy, or psychoeducational resources have a positive impact on emotional symptoms and HRQoL. In contrast to the present study, which highlights negative evolution in the absence of interventions, the findings of Montañés-Masias et al. [68] showed that QoL can improve significantly when emotional and cognitive dimensions are addressed.

Similarly, Broche-Pérez et al. [64] demonstrated that psychological resilience acts as a modulating factor, mitigating the impact of cognitive worry on QoL, particularly in its physical and mental dimensions. This highlights the importance of strengthening personal coping resources from the moment of diagnosis.

Further evidence from recent studies identifies various factors that significantly affect QoL in MS patients. Fatigue, for instance, has been described as one of the most disabling symptoms, with a direct impact on autonomy and emotional well-being. A recent study published by Piñar-Morales et al. [65] confirms that fatigue, anxiety, sleep disorders, depression, and cognitive impairment negatively affect the QoL of MS patients.

### 4.5. Clinical Implications and Future Research Directions

The findings highlight the importance of addressing comorbidities, family history, and lifestyle habits in the comprehensive care of individuals with RRMS. Specific interventions, such as cognitive-behavioral therapy programs, mindfulness sessions, and psychosocial support groups, could significantly improve mental health and social functioning in these patients [59]. For instance, cognitive-behavioral therapy can help identify and modify negative thought patterns that impact QoL perception, while mindfulness can reduce stress and enhance emotional regulation. Psychosocial support groups provide a safe space for sharing experiences and fostering a sense of belonging and community support. These recommendations align with recent studies emphasizing a multidimensional approach to MS management [59]. Moreover, the integration of digital health tools, such as telehealth platforms or mobile applications, could expand the accessibility of these interventions, particularly for patients in remote areas.

The results of Lindberg et al. [63] emphasize the importance of incorporating patient-reported outcome measures into routine clinical practice to monitor disease progression and QoL. Their study highlights that patient-reported outcome measures, such as EQ-5D-3L and MSIS-29, are effective tools for identifying deterioration in QoL, particularly in individuals with PIRA. Similarly, Castillo-Zuñiga et al. [56] emphasize the importance of integrating cognitive assessments, such as Brief International Cognitive Assessment for Multiple Sclerosis, and patient-reported outcomes into routine clinical practice to capture the broader impact of RRMS on patients’ lives. Their findings reinforce the need for multidimensional care approaches that address both cognitive and psychosocial factors. However, the feasibility of implementing these tools in resource-limited settings requires further exploration.

It is important to note that the sample size calculation (35 patients) is based on the population residing in the Lugo health area. Therefore, the results of this study are representative of this specific population and should be interpreted with caution when extrapolated to broader populations. This limitation highlights the need for future studies with larger and more diverse samples to improve the generalizability of the findings and to better understand the broader implications for individuals with RRMS.

The data from Tarasiuk et al. [55] emphasize the importance of reducing disability levels and optimizing DMT access to improve QoL outcomes. Their study underscores the role of tailored therapeutic strategies in addressing the multidimensional needs of MS patients. Meanwhile, Song et al. [57] emphasize the need for integrating HRQoL assessments, such as the EQ-5D-5L, into routine clinical practice to better understand the multifaceted impact of MS on patients’ lives. Their study highlights the critical role of demographic and clinical factors, such as younger age, higher education, and better disability status, in shaping HRQoL outcomes. These findings reinforce the importance of a multidimensional approach to MS management, addressing both clinical and psychosocial factors.

The findings highlight the importance of early and multidimensional interventions targeting emotional and social domains of QoL. For instance, psychological support programs and community-based interventions could mitigate the decline in emotional limitations and social functioning observed over time. Additionally, addressing modifiable lifestyle factors, such as tobacco use, may improve physical limitations and overall QoL. These strategies should be integrated into routine clinical care to enhance long-term outcomes in individuals with RRMS.

Nauta et al. [58] emphasized the need for future randomized trials to establish causal relationships between lifestyle interventions and MS outcomes. Their study demonstrated that participants with higher compliance to the intervention experienced the greatest improvements in QoL, underscoring the importance of personalized and sustained lifestyle adjustments in MS management.

The stability of HRQoL suggests that current treatments may be effective in maintaining overall well-being; however, a more targeted approach is needed to address dimensions that tend to deteriorate, such as emotional limitations and social functioning. Specific intervention designs, such as health education programs focused on self-care and coping strategies, could be implemented to address these areas. For example, educational workshops that include stress management techniques, interpersonal communication skills, and resilience training can empower individuals to face the emotional and social challenges associated with the disease.

Additionally, technology-based interventions, such as mobile applications offering psychoeducational resources and personalized monitoring, could facilitate access to emotional and social support tools from the early stages of the disease. These innovative approaches have the potential to complement traditional care and improve accessibility to support individuals with RRMS.

Future studies should explore the influence of specific factors, such as social support and psychological coping strategies, on the evolution of QoL in these patients. While simpler methods were employed to ensure the reliability and interpretability of results, future studies with larger cohorts should consider the use of these models to better capture temporal changes and individual variability in MSQOL-54 scores. This represents both a methodological limitation and a clinical challenge that warrants further investigation.

## 5. Limitations

This study presents several limitations that must be considered when interpreting the results. First, the small sample size (35 patients) limits the generalizability of the conclusions to a broader population, especially given the heterogeneity of RRMS patients; and limited the applicability of advanced statistical models, such as repeated-measures or mixed-effects designs, which require sufficient statistical power to detect potential effects. Second, the absence of long-term follow-up prevents the evaluation of the evolution of the studied variables beyond the subacute phase, which could provide crucial information on functional and emotional stabilization in later stages. Although the sample size was relatively small, it was consistent with similar studies on RRMS, given the specific inclusion criteria and the prevalence of the disease. Non-parametric tests were employed to account for this limitation and ensure the robustness of the findings Additionally, although multiple clinical and emotional variables were included, other potentially relevant factors, such as social support or access to rehabilitation resources, were not assessed, which may influence the outcomes. The observational design of the study does not allow for definitive causal relationships to be established between the analyzed variables. Finally, the lack of data on access to rehabilitation resources and social support networks represents a limitation, as these factors are known to significantly influence HRQoL in MS populations.

## 6. Conclusions

This study advances the understanding of health-related quality of life (HRQoL) in individuals with relapsing-remitting multiple sclerosis (RRMS) by identifying stable and vulnerable dimensions that require targeted interventions. A key contribution is the recognition of the multifactorial nature of QoL in RRMS, emphasizing the interplay between sociodemographic, clinical, and lifestyle factors. By demonstrating the predictive value of baseline QoL for its subsequent evolution, this research underscores the importance of early and personalized assessments to guide interventions aimed at optimizing outcomes.

Furthermore, the study highlights the critical need to address emotional and social dimensions holistically, as these are particularly vulnerable to deterioration in the early stages of the disease. This calls for the integration of multidimensional, patient-centered approaches into RRMS management, combining physical, psychological, and social strategies to mitigate the disease burden and enhance overall well-being.

By laying the groundwork for future research, this study provides a foundation for developing targeted interventions and improving care strategies for individuals with RRMS. It also underscores the need for larger, more diverse cohorts to validate these findings and to explore the mechanisms underlying the observed vulnerabilities, ultimately contributing to more effective and comprehensive management of RRMS.

## Figures and Tables

**Figure 1 neurolint-17-00195-f001:**
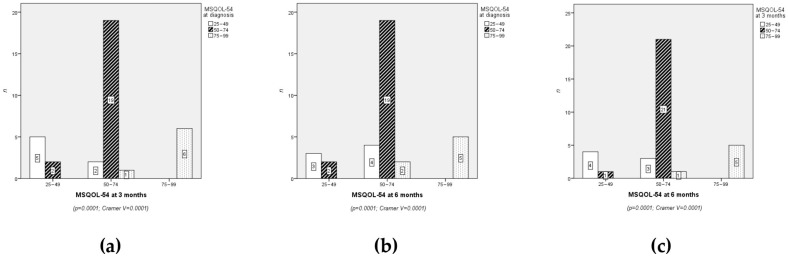
Chi-Square graphical between quality of life at diagnosis, at 3 months and at 6 months. (**a**) Relation between MSQOL-54 at diagnosis and at 3 months. (**b**) Correlation between MSQOL-54 at diagnosis and at 6 months. (**c**) Chi-square between MSQOL-54 at 3 months and at 6 months.

**Figure 2 neurolint-17-00195-f002:**
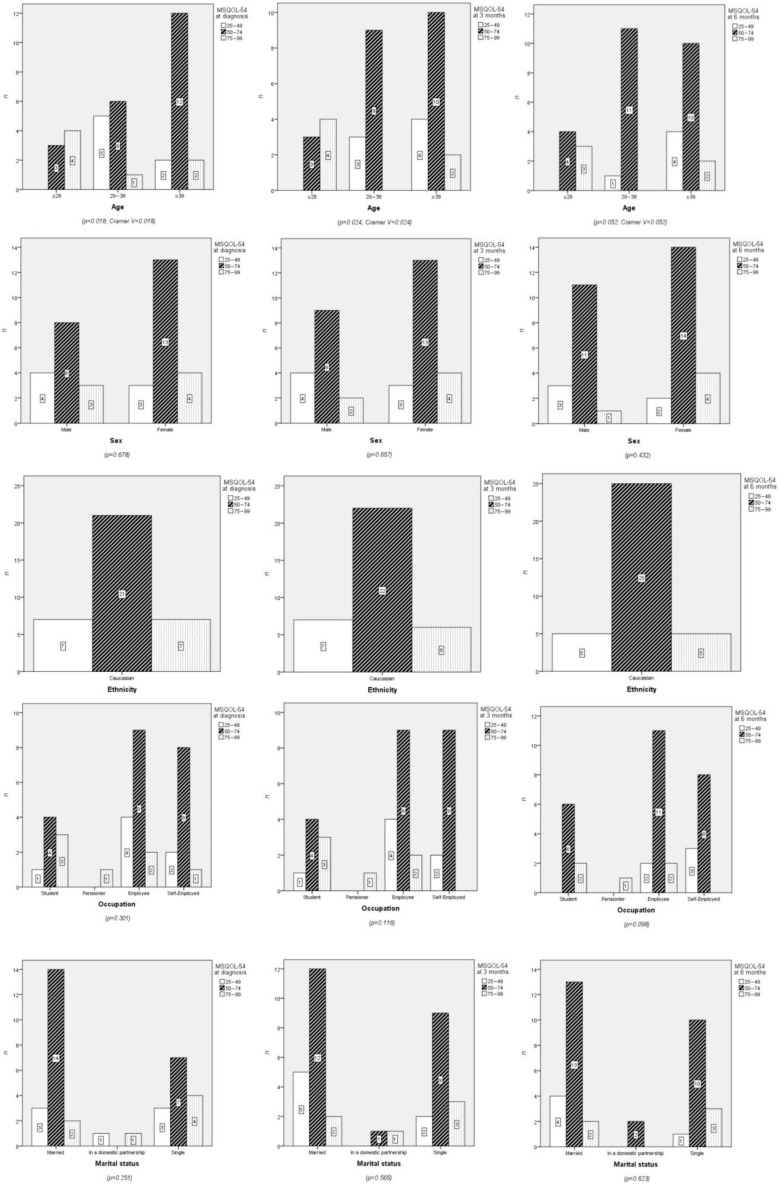
Chi-Square graphical between sociodemographic variables and quality of life at diagnosis, at 3 months and at 6 months.

**Figure 3 neurolint-17-00195-f003:**
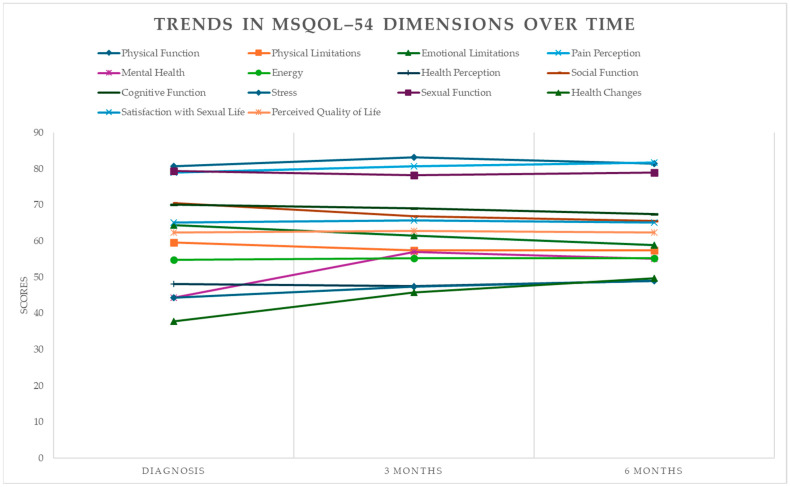
Trends in the QoL dimensions over time.

**Table 1 neurolint-17-00195-t001:** Frequencies and Measures of Central Tendency and Dispersion for the Characteristics of the Studied Variables.

	*n*	%	M	SD	P25	P50	P75	IL	SL
Age			38.29	10.38	30	38	47	19	59
≤28	7	20							
29–38	12	34.3							
≥39	16	45.7							
Sex									
Women	20	57.1							
Men	15	42.9							
Ethnicity									
Caucasian	35	100							
Occupation									
Student	8	22.9							
Pensioner	1	2.9							
Employee	15	42.9							
Self-Employed	11	31.4							
Marital Status									
Married	19	54.3							
In a Domestic Partnership	2	5.7							
Single	14	40							
Educational Level									
High School Diploma	4	11.4							
Intermediate Vocational Training	4	11.4							
Advanced Vocational Training	5	14.3							
Secondary Education	8	22.9							
University Degree	14	40							
Annual Income Level			17,062.86	14,842.21	2200	17,000	27,000	0	60,000
<€12,450	14	40							
€12,450–20,200	7	20							
€20,200–35,200	11	31.4							
€35,200–60,000	2	5.7							
€60,000–300,000	1	2.9							
>€300,000	0	0							
Family History									
No	28	80							
Yes	7	20							
Previous Mononucleosis									
No	33	94.3							
Yes	2	5.7							
Autoimmune Diseases									
No	29	82.9							
Yes	6	17.1							
Pregnancy Planning									
No	32	91.4							
Yes	3	8.6							
Tobacco									
No	27	77.1							
Yes	8	22.9							
Alcohol									
No	32	91.4							
Yes	3	8.6							
Other Substances									
No	34	97.1							
Yes (Cannabis)	1	2.9							
Initial Symptoms									
Visual Disturbance	7	20							
Weakness	4	11.4							
Diplopia	6	17.1							
Hypoesthesia	11	31.4							
Paresthesia	7	20							
Magnetic Resonance Imaging									
Yes	35	100							
Treatment									
Alemtuzumab	1	2.9							
Cladribina	2	5.7							
Corticosteroids	3	8.6							
Diroximel fumarate	2	5.7							
Lemtrada	1	2.9							
Mavenclad	5	14.3							
Ocrelizumab	7	20							
Ocrevus	2	5.7							
Ofatumumab	3	8.6							
Ponesimod	1	2.9							
Tecfidera	1	2.9							
Tysabri	5	14.3							
Ublituximab	1	2.9							
Vumerity	1	2.9							

*n*: sample (*n* = 35); %: frequency; SD: standard deviation; P25. P50. P75: quartiles; IL: inferior limit; SL: superior limit.

**Table 2 neurolint-17-00195-t002:** t-student results for Quality of Life domains between diagnosis and 3–6 months follow-up.

Domain	Diagnosis Measures		3 Months Follow-Up Measures					6 Months Follow-Up Measures				
	Mean	SD	Mean ± SD	Effect Size (d)	t	95% CI (Lower, Upper)	Sig. (Two-Tailed)	Mean ± SD	Effect Size (d)	t	95% CI (Lower, Upper)	Sig. (Two-Tailed)
Physical function	80.67	17.67	83.14 ± 13.13	−0.159	−0.664	(−0.49, 0.17)	0.89	81.52 ± 12.71	−0.055	−0.231	(−0.39, 0.28)	1
Physical limitations	59.57	25.79	57.43 ± 22.17	0.089	0.372	(−0.24, 0.42)	0.51	57.43 ± 21.50	0.090	0.377	(−0.24, 0.42)	0.82
Emotional limitations	64.38	23.03	61.52 ± 24.74	0.120	0.501	(−0.21, 0.45)	0.71	58.86 ± 20.48	0.253	1.060	(−0.08, 0.58)	0.71
Pain	78.93	23.59	80.71 ± 22.40	−0.077	−0.324	(−0.41, 0.25)	0.62	81.79 ± 20.52	−0.129	−0.541	(−0.46, 0.20)	0.29
Mental health	44.34	14.49	57.03 ± 17.17	−0.799	−3.342	(−1.13, −0.47)	0.75	55.09 ± 15.16	−0.725	−3.033	(−1.06, −0.39)	0.59
Energy	54.86	20.27	55.20 ± 18.63	−0.017	−0.073	(−0.35, 0.31)	0.001	55.20 ± 15.91	−0.019	−0.078	(−0.35, 0.31)	0.001
Health Perception	48.11	18.99	47.54 ± 16.77	0.032	0.133	(−0.30, 0.36)	0.94	48.91 ± 17.16	−0.044	−0.185	(−0.38, 0.29)	0.94
Social function	70.48	16.67	66.86 ± 15.55	0.225	0.939	(−0.11, 0.56)	0.89	65.52 ± 12.78	0.334	1.397	(0.00, 0.67)	0.85
Cognitive function	70.12	21.40	69.04 ± 19.78	0.052	0.219	(−0.28, 0.38)	0.35	67.50 ± 19.30	0.129	0.538	(−0.20, 0.46)	0.17
Stress	44.29	22.51	47.38 ± 18.40	−0.150	−0.629	(−0.48, 0.18)	0.83	48.95 ± 17.84	−0.229	−0.960	(−0.56, 0.10)	0.59
Sexual function	79.46	17.12	78.21 ± 18.96	0.069	0.289	(−0.26, 0.40)	0.53	78.93 ± 18.25	0.030	0.125	(−0.30, 0.36)	0.34
Health changes	37.71	18.00	45.71 ± 24.05	−0.377	−1.576	(−0.71, −0.05)	0.77	49.71 ± 23.95	−0.566	−2.370	(−0.90, −0.24)	0.9
Satisfaction with Sexual Life	65.14	23.93	65.71 ± 20.33	−0.026	−0.107	(−0.36, 0.31)	0.12	65.14 ± 20.20	0.000	0.000	(−0.33, 0.33)	0.02
Perceived Quality of Life	62.39	14.15	62.85 ± 12.71	−0.034	−0.143	(−0.37, 0.30)	0.92	62.39 ± 11.77	0.000	0.000	(−0.33, 0.33)	1

*n* = 35, df = 34.

**Table 3 neurolint-17-00195-t003:** t-student results for Quality of Life domains between 3 months and 6 months follow-up.

Domain	Mean ± SD (3 Months)	Mean ± SD (6 Months)	Effect Size (d)	t	95% CI (Lower, Upper)	Sig. (Two-Tailed)
Physical function	83.14 ± 13.13	81.52 ± 12.71	0.125	2.682	(−0.688, 0.939)	0.007
Physical limitations	57.43 ± 22.17	57.43 ± 21.5	0.0	0.407	(−1.374, 1.374)	0.684
Emotional limitations	61.52 ± 24.74	58.86 ± 20.48	0.117	3.847	(−1.416, 1.651)	0.0
Pain	80.71 ± 22.4	81.79 ± 20.52	−0.05	−0.897	(−1.439, 1.338)	0.37
Mental health	57.03 ± 17.17	55.09 ± 15.16	0.12	5.335	(−0.944, 1.184)	0.0
Energy	55.2 ± 18.63	55.2 ± 15.91	0.0	−0.66	(−1.155, 1.155)	0.51
Health Perception	47.54 ± 16.77	48.91 ± 17.16	−0.081	−2.357	(−1.12, 0.959)	0.019
Social function	66.86 ± 15.55	65.52 ± 12.78	0.094	1.567	(−0.87, 1.058)	0.118
Cognitive function	69.04 ± 19.78	67.5 ± 19.3	0.079	2.504	(−1.147, 1.305)	0.012
Stress	47.38 ± 18.4	48.95 ± 17.84	−0.087	−1.61	(−1.227, 1.054)	0.108
Sexual function	78.21 ± 18.96	78.93 ± 18.25	−0.039	−0.406	(−1.214, 1.136)	0.684
Health changes	45.71 ± 24.05	49.71 ± 23.95	−0.167	−4.091	(−1.657, 1.324)	0.0
Satisfaction with Sexual Life	65.71 ± 20.33	65.14 ± 20.2	0.028	−0.024	(−1.232, 1.288)	0.981
Perceived Quality of Life	62.85 ± 12.71	62.39 ± 11.77	0.038	0.639	(−0.75, 0.825)	0.523

*n* = 35; df = 34.

**Table 4 neurolint-17-00195-t004:** Pearson correlations between quality of life at diagnosis and at 3 and 6 months.

		MSQOL-54 at Diagnosis	MSQOL-54 at 3 Months	MSQOL-54 at 6 Months
MSQOL-54 at diagnosis	Correlation Coefficient	1		
	Sig. (two-tailed)			
MSQOL-54 at 3 months	Correlation Coefficient	0.909 **	1	
	Sig. (two-tailed)	0.001		
MSQOL-54 at 6 months	Correlation Coefficient	0.888 **	0.933 **	1
	Sig. (two-tailed)	0.001	0.001	

**. The correlation is significant at the 0.01 level (two-tailed). *n* = 35.

**Table 5 neurolint-17-00195-t005:** t-student results for overall Quality of Life.

Domain	Mean ± SD	Mean Difference	95% CI (Lower, Upper)	t	Sig. (Two-Tailed)	Effect Size (d)
Quality of Life (Diagnosis vs. 3 months)	62.4 ± 14.15 vs. 62.85 ± 12.72	−0.45	(−2.47, 1.58)	−0.448	0.657	0.08
Quality of Life (Diagnosis vs. 6 months)	62.4 ± 14.15 vs. 62.39 ± 11.77	0.01	(−2.24, 2.27)	0.010	0.992	0.00
Quality of Life (3 months vs. 6 months)	62.85 ± 12.72 vs. 62.39 ± 11.77	0.46	(−1.12, 2.03)	0.590	0.559	0.10

*n* = 35; df = 34.

## Data Availability

The datasets generated and/or analyzed during the current study are not publicly available but can be obtained from the corresponding author upon request, subject to privacy and ethical restrictions.

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
