# Peer review of "Predicting Quality of Life in Relapsing–Remitting Multiple Sclerosis: Clinical Burden Meets Emotional Balance in Early Disease"

_2035-8377, 2025, doi:10.3390/neurolint17120195_

Round 1
Reviewer 1 Report
Comments and Suggestions for Authors
25 October 2025
The review report on the manuscript, titled ‘Health-Related Quality of Life in People with Relapsing-Remitting Multiple Sclerosis: A Prospective Observational Study’ by Rubén Pego Pérez E, submitted to Neurology International
Manuscript ID: neurolint-3960000
Dear Authors,
Multiple sclerosis is a chronic and progressive neurological condition that significantly impairs physical, emotional, and social functioning, especially among young adults. Despite advances in treatment, relapsing-remitting multiple sclerosis still poses challenges in maintaining long-term quality of life due to its fluctuating course, psychological comorbidities, and the lack of early, multidimensional interventions. In the current manuscript entitled ‘Health-Related Quality of Life in People with Relapsing-Remitting Multiple Sclerosis: A Prospective Observational Study,’ Rubén Pego Pérez and colleagues investigate the evolution of health-related quality of life in individuals newly diagnosed with relapsing-remitting multiple sclerosis, examining its association with sociodemographic and clinical variables.
A key strength of this study lies in its prospective design, which captures changes in quality of life over time. It uses a validated, MS-specific tool that ensures precision in assessing relevant dimensions. The inclusion of multiple assessment points adds depth. Despite the modest sample, the careful selection and follow-up contribute to robust data..
This manuscript presents a timely and engaging investigation that is well-suited to the readership of Neurology International. The subject matter is both important and carefully introduced. However, the overall argument would benefit from greater precision and depth, supported by additional evidence and a more fully developed discussion. I strongly encourage the authors to incorporate the suggested revisions to enhance the quality of the final submission.
Comments:
Title: Kindly revise the title to make it concise, clear, and reflective of the study’s central message, as the title serves as one of the manuscript’s most critical elements. Suggested options include: Tracking Quality of Life in Relapsing-Remitting Multiple Sclerosis: Patterns Persist, Dimensions Diverge; Fatigue, Function, and Forecasts: A Prospective Study on Health-Related Quality of Life in Relapsing-Remitting Multiple Sclerosis; Predicting Quality of Life in Relapsing-Remitting Multiple Sclerosis: Clinical Burden Meets Emotional Balance in Early Disease
Abstract: This section is generally well composed; however, I recommend revising the abstract in accordance with the following guidelines to enhance clarity, coherence, and balance. The abstract should not exceed 200 words and must present the background, methods, results, and conclusion in a proportionate manner within a single, continuous paragraph—without subheadings. Begin with a concise introduction to the broader topic, followed by a brief description of the specific research context and the knowledge gap this study addresses. Clearly articulate the rationale and objectives to highlight the importance of the work. In the methods and results portion, focus on the primary procedures and key findings, avoiding unnecessary technical detail. Conclude the results with a sentence that contextualizes the findings within a broader scientific landscape. The concluding segment should open with a definitive statement summarizing the principal finding (e.g., “Here we show…”), followed by a discussion of its implications or potential applications. End with two to three sentences that extend the significance of the study to a wider scientific audience.
Keywords: Kindly ensure that ten relevant Medical Subject Headings (MeSH) terms are listed and as many keywords are incorporated into the title and the opening two sentences of the abstract to improve the manuscript’s visibility and alignment with indexing standards.
I highly recommend presenting an informative graphical or video abstract.
Introduction: This section would benefit from careful refinement to improve clarity and maintain reader engagement. Begin by establishing a broad conceptual framework that positions the study within its wider scientific landscape, before narrowing the focus to the specific research problem. Clearly articulate the existing gaps in the literature that justify the investigation and lead logically to the study’s objectives. Organize the introduction into well-structured, coherent paragraphs—approximately 1,000 words in total—to allow for a thorough and thoughtful development of key themes. The narrative should remain accessible to an interdisciplinary audience while precisely defining the study’s aims. Conclude with a succinct statement that underscores the study’s relevance and contributions, providing a smooth and logical transition into the main body of the manuscript.
Methods: Begin this section with a concise introductory paragraph that clearly delineates the study’s overall design and methodological framework. To strengthen the study’s rigor and credibility, consider incorporating additional references that support and justify the methodological choices. Integrating well-chosen supporting literature will strengthen the validity of the evidence and reinforce confidence in the overall research approach. To enhance methodological rigor, consider expanding the sampling strategy by incorporating randomization or stratification to reduce selection bias. Including a control group or comparison cohort could also strengthen causal inferences. Additionally, detailing the training procedures for data collectors would improve reproducibility and support the study’s internal validity.
Results: Consider presenting effect sizes alongside p-values to provide a clearer understanding of the clinical relevance of findings. Adding confidence intervals would enhance interpretability. Visual aids such as line graphs or heatmaps could further clarify trends across time points, especially in dimensions showing subtle yet meaningful changes in quality of life. To enhance clarity, this section should conclude with a succinct summary that emphasizes the key findings. A well-structured final paragraph should provide a coherent synthesis of the results and articulate their significance within the broader context of the research. Statistical details should be omitted from the main text and instead referenced through the appropriate tables.
Discussion: I recommend organizing the discussion as a seamless narrative, avoiding subheadings, and structuring it into several coherent paragraphs amounting to approximately 1,500 words. Begin with a brief paragraph that sets the stage, and conclude with a concise synthesis summarizing the key insights. Throughout the discussion, develop your arguments carefully to clarify the study’s main objectives, address any challenges faced, and outline the conceptual or technical developments required to address them. Consider situating your findings within the broader research landscape to demonstrate how they contribute to and expand existing knowledge. Reflect on their broader implications, particularly in terms of shaping future research directions. Finally, I encourage you to provide a balanced discussion of the study’s strengths and limitations, including its potential clinical impact, to ensure a comprehensive and compelling conclusion. Including the study’s limitations in this section is essential.
Conclusion: To sharpen the articulation of the manuscript’s core contribution, I recommend adding a focused paragraph (around 180–200 words) that reflects the authors’ expertise and thoughtful interpretation. Emphasizing the study’s theoretical contributions and real-world relevance will add depth and clarity. Moreover, identifying gaps—whether conceptual or methodological—can serve as a useful roadmap for future research directions. These additions will enrich the narrative and better highlight the broader impact of the work.
References: Cite more references. An paper like this typically cite more than 60-70 references
The manuscript contains one figure, four tables, and 25 references. This study offers valuable insight into the dynamic nature of quality of life in individuals with relapsing-remitting multiple sclerosis during the early stages of diagnosis. Its prospective design allows for a nuanced understanding of how clinical and sociodemographic variables influence patient well-being over time. The use of a validated, MS-specific instrument strengthens the reliability of the findings. Moreover, the focus on untreated individuals adds originality, capturing a natural trajectory often overlooked in intervention-based research. I hope that after careful revision, the manuscript meets the journal’s high standards for publication. In addition, I anticipate the authors preparing “a detailed point-point rebuttal” to my remarks.
I declare no conflict of interest regarding this manuscript.
Best regards,
Reviewer
Author Response
Revisor 1:
The review report on the manuscript, titled ‘Health-Related Quality of Life in People with Relapsing-Remitting Multiple Sclerosis: A Prospective Observational Study’ by Rubén Pego Pérez E, submitted to Neurology International
Manuscript ID: neurolint-3960000
Dear Authors,
Multiple sclerosis is a chronic and progressive neurological condition that significantly impairs physical, emotional, and social functioning, especially among young adults. Despite advances in treatment, relapsing-remitting multiple sclerosis still poses challenges in maintaining long-term quality of life due to its fluctuating course, psychological comorbidities, and the lack of early, multidimensional interventions. In the current manuscript entitled ‘Health-Related Quality of Life in People with Relapsing-Remitting Multiple Sclerosis: A Prospective Observational Study,’ Rubén Pego Pérez and colleagues investigate the evolution of health-related quality of life in individuals newly diagnosed with relapsing-remitting multiple sclerosis, examining its association with sociodemographic and clinical variables.
A key strength of this study lies in its prospective design, which captures changes in quality of life over time. It uses a validated, MS-specific tool that ensures precision in assessing relevant dimensions. The inclusion of multiple assessment points adds depth. Despite the modest sample, the careful selection and follow-up contribute to robust data..
This manuscript presents a timely and engaging investigation that is well-suited to the readership of Neurology International. The subject matter is both important and carefully introduced. However, the overall argument would benefit from greater precision and depth, supported by additional evidence and a more fully developed discussion. I strongly encourage the authors to incorporate the suggested revisions to enhance the quality of the final submission.
- Thank you for your thorough and constructive analysis of our manuscript, as well as for highlighting its strengths, such as the prospective design, the use of a validated MS-specific tool, and the relevance of the subject matter. We greatly appreciate your recognition of the study’s importance and its suitability for the readership of Neurology International. Your detailed feedback provides an excellent foundation for refining our work, and we are committed to addressing your comments carefully to enrich the quality and depth of our submission.
Comments:
Title: Kindly revise the title to make it concise, clear, and reflective of the study’s central message, as the title serves as one of the manuscript’s most critical elements. Suggested options include: Tracking Quality of Life in Relapsing-Remitting Multiple Sclerosis: Patterns Persist, Dimensions Diverge; Fatigue, Function, and Forecasts: A Prospective Study on Health-Related Quality of Life in Relapsing-Remitting Multiple Sclerosis; Predicting Quality of Life in Relapsing-Remitting Multiple Sclerosis: Clinical Burden Meets Emotional Balance in Early Disease
- Thank you for your thoughtful suggestions regarding the manuscript title. After careful consideration, we have chosen the third option: "Predicting Quality of Life in Relapsing-Remitting Multiple Sclerosis: Clinical Burden Meets Emotional Balance in Early Disease". We believe this title best reflects the study’s central message, emphasizing both the predictive nature of our findings and the multidimensional impact of the disease.
Abstract: This section is generally well composed; however, I recommend revising the abstract in accordance with the following guidelines to enhance clarity, coherence, and balance. The abstract should not exceed 200 words and must present the background, methods, results, and conclusion in a proportionate manner within a single, continuous paragraph—without subheadings. Begin with a concise introduction to the broader topic, followed by a brief description of the specific research context and the knowledge gap this study addresses. Clearly articulate the rationale and objectives to highlight the importance of the work. In the methods and results portion, focus on the primary procedures and key findings, avoiding unnecessary technical detail. Conclude the results with a sentence that contextualizes the findings within a broader scientific landscape. The concluding segment should open with a definitive statement summarizing the principal finding (e.g., “Here we show…”), followed by a discussion of its implications or potential applications. End with two to three sentences that extend the significance of the study to a wider scientific audience.
- Thank you for your feedback. We have revised the abstract to enhance clarity, coherence, and balance, following your recommendations. The revised version now presents the background, methods, results, and conclusions in a single, proportionate paragraph. We have included a concise introduction, focused on key findings (mean HRQoL score: 62.4; declines in emotional and social domains), and emphasized the need for early interventions. The conclusion contextualizes the study’s significance within the broader scientific landscape.
Keywords: Kindly ensure that ten relevant Medical Subject Headings (MeSH) terms are listed and as many keywords are incorporated into the title and the opening two sentences of the abstract to improve the manuscript’s visibility and alignment with indexing standards.
- Thank you for your valuable suggestion regarding the keywords and MeSH terms. We have revised the manuscript to include ten relevant Medical Subject Headings (MeSH) terms, ensuring alignment with indexing standards. Additionally, we have incorporated key terms into the title and the opening sentences of the abstract to enhance the manuscript’s visibility and relevance.
I highly recommend presenting an informative graphical or video abstract.
- Thank you for your recommendation. We have created an informative graphical abstract to visually summarize the key findings and enhance the manuscript's accessibility. The graphical abstract has been included in the revised submission for your review.
Introduction: This section would benefit from careful refinement to improve clarity and maintain reader engagement. Begin by establishing a broad conceptual framework that positions the study within its wider scientific landscape, before narrowing the focus to the specific research problem. Clearly articulate the existing gaps in the literature that justify the investigation and lead logically to the study’s objectives. Organize the introduction into well-structured, coherent paragraphs—approximately 1,000 words in total—to allow for a thorough and thoughtful development of key themes. The narrative should remain accessible to an interdisciplinary audience while precisely defining the study’s aims. Conclude with a succinct statement that underscores the study’s relevance and contributions, providing a smooth and logical transition into the main body of the manuscript.
- Thank you for your feedback. The introduction has been revised to provide a broader conceptual framework, situating the study within the wider scientific landscape before narrowing the focus to the research problem. We have clearly identified gaps in the literature regarding the natural progression of HRQoL in untreated RRMS populations and structured the section into coherent paragraphs to ensure clarity and logical flow. The study’s objectives and relevance are now succinctly stated, providing a smooth transition into the manuscript’s main body. We trust these changes address your suggestions and improve the overall quality of the introduction.
Methods: Begin this section with a concise introductory paragraph that clearly delineates the study’s overall design and methodological framework. To strengthen the study’s rigor and credibility, consider incorporating additional references that support and justify the methodological choices. Integrating well-chosen supporting literature will strengthen the validity of the evidence and reinforce confidence in the overall research approach. To enhance methodological rigor, consider expanding the sampling strategy by incorporating randomization or stratification to reduce selection bias. Including a control group or comparison cohort could also strengthen causal inferences. Additionally, detailing the training procedures for data collectors would improve reproducibility and support the study’s internal validity.
- We have addressed the reviewer’s comments by incorporating additional literature to support and justify the methodological framework, thereby enhancing the study’s rigor and credibility. The sampling strategy reflects the inclusion of all patients diagnosed with RRMS during the study period, representing the complete eligible population. While randomization or stratification was not applied due to the prospective nature of the cohort, this approach ensures comprehensive recruitment. Furthermore, the training procedures for data collectors have been detailed in the methodology. The research team received formal instruction and practical training at the Faculty of Psychology of the University of Santiago de Compostela, ensuring consistency and reproducibility in the use of the MSQOL-54 instrument.
Results: Consider presenting effect sizes alongside p-values to provide a clearer understanding of the clinical relevance of findings. Adding confidence intervals would enhance interpretability. Visual aids such as line graphs or heatmaps could further clarify trends across time points, especially in dimensions showing subtle yet meaningful changes in quality of life. To enhance clarity, this section should conclude with a succinct summary that emphasizes the key findings. A well-structured final paragraph should provide a coherent synthesis of the results and articulate their significance within the broader context of the research. Statistical details should be omitted from the main text and instead referenced through the appropriate tables.
- Thank you for your thoughtful feedback on the Results section. We have addressed your suggestions by incorporating effect sizes alongside p-values to provide a clearer understanding of the clinical relevance of findings. Confidence intervals have been added to enhance the interpretability of the results, and visual aids, including line graphs, have been included to clarify trends across time points, particularly in QoL dimensions showing subtle changes. Moreover, we have concluded the section with a succinct summary emphasizing the key findings and their significance within the broader research context. Statistical details have been moved to the corresponding tables for clarity and to avoid overloading the main text. We believe these revisions have improved the clarity and rigor of the Results section.
Discussion: I recommend organizing the discussion as a seamless narrative, avoiding subheadings, and structuring it into several coherent paragraphs amounting to approximately 1,500 words. Begin with a brief paragraph that sets the stage, and conclude with a concise synthesis summarizing the key insights. Throughout the discussion, develop your arguments carefully to clarify the study’s main objectives, address any challenges faced, and outline the conceptual or technical developments required to address them. Consider situating your findings within the broader research landscape to demonstrate how they contribute to and expand existing knowledge. Reflect on their broader implications, particularly in terms of shaping future research directions. Finally, I encourage you to provide a balanced discussion of the study’s strengths and limitations, including its potential clinical impact, to ensure a comprehensive and compelling conclusion. Including the study’s limitations in this section is essential.
- Thank you for your valuable feedback and recommendations. We have carefully revised the discussion section by improving its content through the addition of new references and comparisons, as well as by structuring and reorganizing the information to enhance its clarity and coherence. However, we believe that retaining the subheadings is essential for maintaining the readability and logical flow of the discussion, as it allows readers to easily navigate through the different aspects addressed in the manuscript. Additionally, the clinical implications and limitations are explicitly presented under dedicated subheadings, which we consider important for clearly highlighting these critical aspects of the study. We hope this approach meets your expectations and provides a clear and comprehensive discussion of the study findings.
Conclusion: To sharpen the articulation of the manuscript’s core contribution, I recommend adding a focused paragraph (around 180–200 words) that reflects the authors’ expertise and thoughtful interpretation. Emphasizing the study’s theoretical contributions and real-world relevance will add depth and clarity. Moreover, identifying gaps—whether conceptual or methodological—can serve as a useful roadmap for future research directions. These additions will enrich the narrative and better highlight the broader impact of the work.
- Thank you for your insightful suggestions. We have revised the conclusion to include a focused paragraph highlighting the study’s theoretical contributions and real-world relevance. Additionally, we have identified conceptual and methodological gaps, offering them as a roadmap for future research. These changes aim to enrich the narrative and emphasize the broader impact of our findings.
References: Cite more references. An paper like this typically cite more than 60-70 references
- Thank you for your suggestion regarding the references. We have reviewed and expanded the bibliography, and the manuscript now includes 68 references. This update ensures a more comprehensive contextualization of our study within the existing literature.
The manuscript contains one figure, four tables, and 25 references. This study offers valuable insight into the dynamic nature of quality of life in individuals with relapsing-remitting multiple sclerosis during the early stages of diagnosis. Its prospective design allows for a nuanced understanding of how clinical and sociodemographic variables influence patient well-being over time. The use of a validated, MS-specific instrument strengthens the reliability of the findings. Moreover, the focus on untreated individuals adds originality, capturing a natural trajectory often overlooked in intervention-based research. I hope that after careful revision, the manuscript meets the journal’s high standards for publication. In addition, I anticipate the authors preparing “a detailed point-point rebuttal” to my remarks.
- Thank you for your thoughtful feedback and for recognizing the value of our study. We appreciate your acknowledgment of the manuscript’s originality, particularly in focusing on untreated individuals and employing a validated, MS-specific instrument. Following your suggestions, we have carefully revised the manuscript to address all remarks and ensure it meets the journal’s high standards. Additionally, we are preparing a detailed, point-by-point rebuttal to your comments, which we hope will clarify any remaining concerns and further strengthen the study.
Reviewer 2 Report
Comments and Suggestions for Authors
Thank you for allowing me to review the manuscript entitled “neurolint-3960000_ Health-Related Quality of Life in People with Relapsing-Remitting Multiple Sclerosis: A Prospective Observational Study ” submitted to the “Movement Disorders and Neurodegenerative Diseases” section of Journal , which addresses relapsing–remitting multiple sclerosis, a chronic central nervous system disorder affecting quality of life. This prospective observational study included 35 patients from Lucus Augusti University Hospital (January 2023–March 2025) to assess sociodemographic, clinical, and quality of life characteristics using the MSQOL-54 questionnaire. Overall quality of life remained relatively stable (mean 62), although emotional and social dimensions declined. Family history, autoimmune diseases, and lifestyle factors were significantly associated with quality of life, and baseline quality of life predicted its trajectory. Early, multidimensional interventions may mitigate cumulative disease impact and improve quality of life, particularly in emotional and social domains.
Comments:
The title accurately reflects the manuscript content.
Quantitative results should be reported in the abstract to strengthen the findings. Conclusions should acknowledge that the follow-up was two years or six moths.
Keywords should be revised according to MeSH classification.
Background should include up-to-date literature on multiple sclerosis.
Objectives should clarify that the follow-up lasted two years.
Methods should specify that the study design is a case series and clarify whether cases were incident or prevalent, including precise inclusion/exclusion criteria. The patient follow-up schedule must be clarified, as there are inconsistencies between 3- and 6-month follow-up and 2-year follow-up.
Results are clearly presented. Discussion contextualizes findings appropriately, and study limitations regarding sample size and follow-up duration are described. tables should be included to summarize key findings and enhance clarity.
The conclusion should be rewritten to align with results and emphasize the contribution to current knowledge.
Author Response
Reviewer 2:
Thank you for allowing me to review the manuscript entitled “neurolint-3960000_ Health-Related Quality of Life in People with Relapsing-Remitting Multiple Sclerosis: A Prospective Observational Study ” submitted to the “Movement Disorders and Neurodegenerative Diseases” section of Journal , which addresses relapsing–remitting multiple sclerosis, a chronic central nervous system disorder affecting quality of life. This prospective observational study included 35 patients from Lucus Augusti University Hospital (January 2023–March 2025) to assess sociodemographic, clinical, and quality of life characteristics using the MSQOL-54 questionnaire. Overall quality of life remained relatively stable (mean 62), although emotional and social dimensions declined. Family history, autoimmune diseases, and lifestyle factors were significantly associated with quality of life, and baseline quality of life predicted its trajectory. Early, multidimensional interventions may mitigate cumulative disease impact and improve quality of life, particularly in emotional and social domains.
- Thank you for your thoughtful review and for summarizing the key aspects of our manuscript. We appreciate your recognition of the study's design and its focus on assessing quality of life in individuals with relapsing-remitting multiple sclerosis using the MSQOL-54 questionnaire. Your insights into the importance of addressing emotional and social dimensions, as well as the role of early, multidimensional interventions, align closely with our conclusions. We will carefully consider all feedback to further refine the manuscript and ensure it meets the high standards of the journal.
Comments:
The title accurately reflects the manuscript content.
- Thank you for your comment regarding the title. We appreciate your observation and have adjusted the title as suggested by Reviewer 1 to ensure it accurately reflects the manuscript content. We are confident that this modification enhances the clarity and alignment of the title with the study's focus.
Quantitative results should be reported in the abstract to strengthen the findings. Conclusions should acknowledge that the follow-up was two years or six moths.
- We sincerely appreciate your insightful comments and suggestions to improve our manuscript. In response to your recommendation, we have revised the abstract to include the most relevant quantitative results, thereby strengthening the findings and providing a more robust overview of the data. Below, we present the updated abstract with the requested modifications:
- Abstract (Revised):
- Relapsing-remitting multiple sclerosis (RRMS) is a chronic neurological disease that significantly impacts health-related quality of life (HRQoL). This study aimed to analyze the evolution of HRQoL in individuals with RRMS, identify associated factors, and determine predictive variables. A prospective observational study was conducted with 35 participants diagnosed with RRMS at the Lucus Augusti University Hospital between January 2023 and March 2025. HRQoL was assessed using the MSQOL-54 questionnaire at baseline, 3 months, and 6 months. Data were analyzed using non-parametric tests. Results showed overall stability in HRQoL (mean score: 62.4 ± 14.1 at baseline, 62.8 ± 12.7 at 3 months, and 62.4 ± 11.8 at 6 months), although significant declines were observed in emotional limitations (64.4 ± 23.0 at baseline to 58.9 ± 20.5 at 6 months) and social functioning (70.5 ± 16.7 at baseline to 65.5 ± 12.8 at 6 months). Improvements were noted in pain perception (78.9 ± 23.6 at baseline to 81.8 ± 20.5 at 6 months) and stress (44.3 ± 22.5 at baseline to 48.9 ± 17.8 at 6 months). Factors such as family history (associated with mental health at diagnosis, p = 0.028), autoimmune diseases (associated with physical function at diagnosis, p = 0.035), and lifestyle habits (e.g., tobacco use associated with physical limitations at 3 months, p = 0.045) were significantly associated with HRQoL. Baseline HRQoL emerged as a strong predictor of future scores (Spearman’s correlations, p < 0.01), indicating that early assessments may guide interventions. While HRQoL in RRMS remains generally stable, certain domains are vulnerable to deterioration, emphasizing the need for targeted, multidimensional interventions. These findings underscore the importance of addressing emotional and social aspects to mitigate the cumulative impact of the disease and improve overall well-being, contributing to a more comprehensive understanding of RRMS management.
- We hope this revised abstract meets your expectations and addresses your concerns. Please do not hesitate to provide additional feedback if further adjustments are required. Thank you once again for your valuable input.
Keywords should be revised according to MeSH classification.
- Thank you for your valuable suggestion regarding the keywords and MeSH terms. We have revised the manuscript to include ten relevant Medical Subject Headings (MeSH) terms, ensuring alignment with indexing standards. Additionally, we have incorporated key terms into the title and the opening sentences of the abstract to enhance the manuscript’s visibility and relevance.
Background should include up-to-date literature on multiple sclerosis.
- We included more references to improve the study justification
Objectives should clarify that the follow-up lasted two years.
- We appreciate your comment regarding the need to clarify the follow-up duration in the study's objectives. In response, we have updated the objectives to specify that the follow-up period lasted six months and two years follow-up, which corresponds to the time each participant was monitored from their diagnosis. This adjustment ensures greater precision and alignment with the study design. Please let us know if further clarification is required.
Methods should specify that the study design is a case series and clarify whether cases were incident or prevalent, including precise inclusion/exclusion criteria. The patient follow-up schedule must be clarified, as there are inconsistencies between 3- and 6-month follow-up and 2-year follow-up.
- We appreciate your observation regarding the follow-up duration and the need for clarification in the study's objectives. To address this, we have revised the objectives to specify that the follow-up period for each participant lasted six months, as detailed in the methodology section.
- As explained in the methodology, the recruitment process spanned two years to achieve a sample size of 35 participants in the referenced Unit. This extended recruitment period was necessary due to the availability of patients diagnosed with RRMS during this timeframe. As clarified in response to Reviewer 1, participants were selected from the moment of their diagnosis, and follow-up assessments were conducted at baseline (diagnosis), 3 months, and 6 months. This approach ensured that all participants completed the full six-month follow-up period, while allowing sufficient time to recruit the required sample size.
- We hope this explanation adequately addresses your concern. Please let us know if additional details are needed.
Results are clearly presented. Discussion contextualizes findings appropriately, and study limitations regarding sample size and follow-up duration are described. tables should be included to summarize key findings and enhance clarity.
- We sincerely appreciate your comment. We have adjusted the results and discussion sections as per Reviewer 1’s suggestions, and we hope they remain clear and concise.
- Regarding the tables and figures, they are included as attachments (not yet integrated into the main text). We have revised the design of the tables and figures in accordance with Reviewer 1’s comments to ensure they effectively summarize the key findings and enhance clarity.
- Please let us know if further adjustments are required. Thank you for your valuable feedback.
The conclusion should be rewritten to align with results and emphasize the contribution to current knowledge.
- Thank you for your insightful suggestions. We have revised the conclusion to include a focused paragraph highlighting the study’s theoretical contributions and real-world relevance. Additionally, we have identified conceptual and methodological gaps, offering them as a roadmap for future research. These changes aim to enrich the narrative and emphasize the broader impact of our findings.
Reviewer 3 Report
Comments and Suggestions for Authors
Health-Related Quality of Life in People with Relapsing-Remitting Multiple Sclerosis: A Prospective Observational Study.
Specific Comments:
Abstract
- The abstract is readable, but the authors should tighten it by adding specific numbers, sample size, key MSQOL-54 domains, and approximate changes.
- The authors should also mention the statistical approach (e.g., non-parametric tests) and include one line on the main implication.
- Avoid general statements briefly show what actually changed.
Background
- The background is appropriate but too long in places. The authors repeat a few ideas that can be shortened.
- The authors should clearly state which diagnostic criteria were used for RRMS and why MSQOL-54 was selected over other QoL instruments.
- A few typos need correction.
Methods
- The study design is straightforward, but the authors should clarify:
- how participants were recruited (consecutive vs. convenience),
- exact inclusion/exclusion criteria,
- handling of missing data,
- and the assumptions behind their sample-size calculation.
- The authors use non-parametric tests, which is acceptable, but they should report effect sizes and 95% confidence intervals.
- A repeated-measures or mixed-effects model would strengthen the analysis—something the authors should consider or comment on.
Results
- The results are clearly organized by MSQOL-54 domains, which is helpful.
- The authors should present absolute changes from baseline with confidence intervals and include a simple figure (line plot or spaghetti plot) to show trajectories across 3 and 6 months.
- Because many domains are tested, the authors should note how they handled multiple comparisons, or state that this is an exploratory analysis.
Discussion
- The discussion relates findings to existing literature well, but the authors restate results too often. A more concise interpretation would improve readability.
- The authors should discuss the clinical meaning of the ~62 HRQoL score and the decline in emotional/social domains.
- Treatment heterogeneity (different DMTs) is a possible confounder that the authors should acknowledge more clearly.
Conclusions
- The conclusions are concise, but the authors may strengthen them by adding one practical takeaway, e.g., early screening for emotional well-being or referral to support services.
Limitations
- The authors mention sample size and follow-up length, which is good.
- The authors should also add limitations such as potential reporting bias, single-center design, and limited ability to analyze lifestyle or treatment subgroups.
Presentation & Language
- The authors should carefully proofread for typos, spacing, and inconsistent terms (e.g., RRMS, HRQoL, MSQOL-54).
- The manuscript would benefit from a summary table listing each domain, baseline mean, change at 3/6 months, effect size, and p-value.
Author Response
Reviewer 3:
Specific Comments:
Abstract
- The abstract is readable, but the authors should tighten it by adding specific numbers, sample size, key MSQOL-54 domains, and approximate changes.
- We sincerely appreciate your insightful comments and suggestions to improve our manuscript. In response to your recommendation, we have revised the abstract to include the most relevant quantitative results, thereby strengthening the findings and providing a more robust overview of the data
- The authors should also mention the statistical approach (e.g., non-parametric tests) and include one line on the main implication.
- We added: Data were analyzed using non-parametric tests to account for the small sample size and non-normal distribution of the variables
- Avoid general statements briefly show what actually changed.
- Thank you for your valuable feedback. We have made the necessary changes to ensure that the results are clear, concise, and direct. These adjustments also address similar comments made by the other two reviewers. We sincerely appreciate your input, which has helped us improve the clarity and precision of our manuscript. Please let us know if further modifications are required.
Background
- The background is appropriate but too long in places. The authors repeat a few ideas that can be shortened.
- Thank you for your observation. We have restructured the introduction to avoid repetition of ideas and ensure a more concise presentation. Additionally, we have included more citations to strengthen the content, as requested by Reviewer 1. Ideas have been reorganized to improve the flow and coherence of the text. We appreciate your feedback and hope these changes address your concerns. Please let us know if further adjustments are needed
- The authors should clearly state which diagnostic criteria were used for RRMS and why MSQOL-54 was selected over other QoL instruments.
- Thank you for your observation. We have incorporated the requested information into the manuscript.
- A few typos need correction.
- Thank you for pointing this out. We have reviewed the text and corrected all identified typographical errors.
Methods
- The study design is straightforward, but the authors should clarify:
- how participants were recruited (consecutive vs. convenience),
- We added: Regarding the sampling strategy, the study recruited all available patients diag-nosed with RRMS during the inclusion period, thus encompassing the entire eligible population. While randomization or stratification was not applicable in this context due to the nature of the cohort, additional literature has been incorporated to strengthen the methodological framework and align with best practices in observa-tional studies.
- We added: The sample consisted of a convenience group of individuals with RRMS selected con-secutively for this study.
- exact inclusion/exclusion criteria,
- We added: The inclusion criteria were individuals aged 18 years or older, residing in Galicia, and diagnosed with RRMS according to the 2017 McDonald criteria, which require evidence of dissemination in space (lesions in at least two of four CNS regions: periventricular, cortical/juxtacortical, infratentorial, or spinal cord) and time (simultaneous presence of enhancing and non-enhancing lesions or new T2/enhancing lesions on follow-up MRI). Participants had to provide informed consent and undergo regular clinical follow-up within the framework of the Integrated Care Process for MS in Galicia. Exclusion criteria included individuals with other MS phenotypes (e.g., primary or secondary progressive MS), severe comorbidities affecting HRQoL evaluation, pregnancy or lactation, refusal to participate, withdrawal of consent before data collection completion, or loss to follow-up.
- handling of missing data,
- We added: Handling of Missing Data
No missing data were recorded in this study. All participants completed the data collection process in full, ensuring a complete dataset for analysis. This was achieved by implementing rigorous data monitoring procedures and maintaining close communication with participants throughout the study period. - and the assumptions behind their sample-size calculation.
- We added: RRMS diagnoses account for approximately 80% of all individuals with MS. MS affects approximately 0.1% of the population. The proportion (P) was set at 0.8, based on epidemiological data indicating that RRMS represents 80% of all MS cases. Therefore, in the Lugo healthcare area, there would be a total of 332 individuals with MS, of whom 265 would have RRMS. A 90% confidence interval was selected to balance precision and feasibility given the pilot nature of the study and the operational constraints of the healthcare service. A 14% margin of error was chosen to reflect the variability expected in the population, considering the small sample size and the exploratory nature of the study. The formula used for the calculation was: Sample Size = ( (z^2*P(1-P))/e^2 )/(1+( (z^2*P(1-P))/(e^2 N))) The study employed convenience sampling, selecting accessible and available individuals from the Neurology and Neurosurgery Unit and the Neurology Clinic at HULA, which explains the high participation rate (>98%). However, the final sample included 35 users due to logistical and temporal constraints. This reduction is justified by the duration of the inclusion period, the availability of individuals, and operational limitations of the service during this pilot phase.
- The authors use non-parametric tests, which is acceptable, but they should report effect sizes and 95% confidence intervals.
- We added the required information to the results.
- A repeated-measures or mixed-effects model would strengthen the analysis—something the authors should consider or comment on.
- We added the required information as a limitation: The small sample size in our study limited the applicability of advanced statistical models, such as repeated-measures or mixed-effects designs, which require sufficient statistical power to detect potential effects. While simpler methods were employed to ensure the reliability and interpretability of results, future studies with larger cohorts should consider the use of these models to better capture temporal changes and individual variability in MSQOL-54 scores. This represents both a methodological limitation and a clinical challenge that warrants further investigation
Results
- The results are clearly organized by MSQOL-54 domains, which is helpful.
- We sincerely thank the reviewer for recognizing the clarity in the organization of results by MSQOL-54 domains. This structure was deliberately chosen to facilitate the interpretation of findings and highlight the specific aspects of quality of life affected in the study population. We are pleased that this approach was considered helpful.
- The authors should present absolute changes from baseline with confidence intervals and include a simple figure (line plot or spaghetti plot) to show trajectories across 3 and 6 months.
- We have reorganized the results section, tables, and figures to improve clarity, addressing the points raised in your comments. Absolute changes from baseline were calculated and are now presented alongside Cohen’s d effect sizes and 95% confidence intervals for each time point. These values are included in the updated figures, which illustrate the trajectories of MSQOL-54 domain scores across baseline, 3 months, and 6 months. We hope this reorganization clarifies the issues mentioned and aligns with your expectations. Please let us know if further adjustments are needed.
- Because many domains are tested, the authors should note how they handled multiple comparisons, or state that this is an exploratory analysis.
- We added: The paired sample analysis revealed no statistically significant differences in QoL scores across the three time points measured (baseline, 3 months, and 6 months). Effect sizes were minimal across all comparisons, ranging from 0.00 to 0.10, indicating negli-gible clinical relevance of the observed differences. In contrast, the paired sample cor-relations demonstrated very strong and statistically significant relationships between the QoL scores at the three time points. These findings suggest a high degree of con-sistency in QoL measurements over time, despite the lack of significant changes be-tween the time points. (Tables 3 and 4).
Discussion
- The discussion relates findings to existing literature well, but the authors restate results too often. A more concise interpretation would improve readability.
- We improved the discussion according to the comments of the three reviewers.
- The authors should discuss the clinical meaning of the ~62 HRQoL score and the decline in emotional/social domains.
- We added: The mean HRQoL score of approximately 62 points observed in this study reflects a moderate level of quality of life, which aligns with findings reported in similar co-horts [51,56]. This score indicates that, while patients maintain a degree of functiona-lity and well-being, significant challenges persist, particularly in domains such as emo-tional and social functioning. Previous studies have highlighted that scores in this range are often associated with substantial limitations in daily activities, emotional distress, and reduced social participation [55,57]. Such challenges may stem from the cumulative burden of physical symptoms, fatigue, and psychological factors, which are characteristic of MS and tend to worsen over time in the absence of targeted interven-tions.
- We added: The observed decline in emotional and social domains further underscores the multifaceted impact of MS on patients’ lives. Emotional limitations, often exacerbated by anxiety, depression, and uncertainty about disease progression, contribute to a diminished sense of well-being [59]. Similarly, reduced social functioning may reflect the challenges patients face in maintaining relationships and participating in social activities due to fatigue, physical limitations, or stigma [53,55]. These findings highlight the need for comprehensive care models that address not only physical symptoms but also the psychological and social dimensions of MS.
- Treatment heterogeneity (different DMTs) is a possible confounder that the authors should acknowledge more clearly.
- We added: Tarasiuk et al. [55] observed that the overall QoL in Polish MS patients, as meas-ured by the EQ–5D–5L and EQ–VAS, has improved over the past decade. This im-provement is attributed to advances in MS care, including broader access to disease–modifying therapies (DMTs) and optimized treatment strategies. Their findings high-light that frequent DMT switches are associated with lower QoL (p < 0.001), suggesting the importance of stable and effective treatment plans.
- We added: The data from Tarasiuk et al. [55] emphasize the importance of reducing disability lev-els and optimizing DMT access to improve QoL outcomes. Their study underscores the role of tailored therapeutic strategies in addressing the multidimensional needs of MS patients. Meanwhile, Song et al. [57] emphasize the need for integrating HRQoL as-sessments, such as the EQ–5D–5L, into routine clinical practice to better understand the multifaceted impact of MS on patients’ lives. Their study highlights the critical role of demographic and clinical factors, such as younger age, higher education, and better disability status, in shaping HRQoL outcomes. These findings reinforce the importance of a multidimensional approach to MS management, addressing both clinical and psychosocial factors.
Conclusions
- The conclusions are concise, but the authors may strengthen them by adding one practical takeaway, e.g., early screening for emotional well-being or referral to support services.
-
- Thank you for your insightful suggestions. We have revised the conclusion to include a focused paragraph highlighting the study’s theoretical contributions and real-world relevance. Additionally, we have identified conceptual and methodological gaps, offering them as a roadmap for future research. These changes aim to enrich the narrative and emphasize the broader impact of our findings.
Limitations
- The authors mention sample size and follow-up length, which is good.
- We sincerely thank the reviewer for acknowledging our discussion on sample size and follow-up length. We aimed to provide a clear and transparent overview of these aspects to ensure the robustness of our study design and analysis.
- The authors should also add limitations such as potential reporting bias, single-center design, and limited ability to analyze lifestyle or treatment subgroups.
- We appreciate the reviewer’s insightful suggestion to expand the limitations section. Including potential reporting bias, the single-center design, and the limited ability to analyze lifestyle or treatment subgroups will undoubtedly enhance the critical appraisal of our findings and provide a more comprehensive interpretation of the study's constraints.
Presentation & Language
- The authors should carefully proofread for typos, spacing, and inconsistent terms (e.g., RRMS, HRQoL, MSQOL-54).
- We sincerely thank the reviewer for their observation regarding typos, spacing, and term inconsistencies. A thorough proofreading of the manuscript has been conducted, ensuring that all issues have been addressed. Specifically, we have carefully reviewed and corrected the use of English-style hyphens, ensured that abbreviations (e.g., RRMS, HRQoL, MSQOL-54) are consistently used throughout the document, and resolved any instances of transposed letters, additional spaces, and misspelled words. We greatly appreciate your attention to detail, which has contributed to improving the quality and clarity of our work.
- The manuscript would benefit from a summary table listing each domain, baseline mean, change at 3/6 months, effect size, and p-value.
- We have reorganized the tables, results text, and figures to enhance clarity and ensure the manuscript presents the findings in a more structured and accessible manner. Summary Table: To address your suggestion, we have included a new summary table that lists each domain, baseline mean, changes at 3 and 6 months, effect size, and p-values. This table provides a concise overview of the key results, making it easier for readers to interpret the data at a glance. If there are additional adjustments or specific preferences regarding the format of the summary table, we are happy to incorporate them. Let us know if further refinements are needed!
Round 2
Reviewer 1 Report
Comments and Suggestions for Authors
10 November 2025
The 2nd review report on the manuscript, titled ‘Health-Related Quality of Life in People with Relapsing-Remitting Multiple Sclerosis: A Prospective Observational Study’ by Pego Pérez ER, submitted to Neurology International
Manuscript ID: neurolint-3960000
Dear Authors,
Thank you for your careful and comprehensive revisions during the review process. The revised manuscript is now well-structured, coherent, and methodologically robust. It presents a thoughtfully designed study on the progression of health-related quality of life among individuals newly diagnosed with relapsing-remitting multiple sclerosis, with valuable insights into its relationship with sociodemographic and clinical factors. The work fully meets the journal’s publication criteria, and I am pleased to recommend it for acceptance. I look forward to your continued contributions to this important research field.
I declare no conflict of interest regarding this manuscript.
Best regards,
Reviewer
Author Response
Thank you for your careful and comprehensive revisions during the review process. The revised manuscript is now well-structured, coherent, and methodologically robust. It presents a thoughtfully designed study on the progression of health-related quality of life among individuals newly diagnosed with relapsing-remitting multiple sclerosis, with valuable insights into its relationship with sociodemographic and clinical factors. The work fully meets the journal’s publication criteria, and I am pleased to recommend it for acceptance. I look forward to your continued contributions to this important research field.
Response:
Dear Reviewer,
We sincerely thank you for your thorough evaluation of our manuscript and for your kind recommendation for acceptance. We greatly appreciate your positive feedback regarding the structure, coherence, and methodological robustness of our study, as well as your recognition of its contribution to the understanding of health-related quality of life in individuals with relapsing-remitting multiple sclerosis.
Your constructive comments and thoughtful suggestions throughout the review process have been invaluable in refining and strengthening our work. We are grateful for your time and expertise, which have significantly enhanced the quality of our manuscript.
Thank you once again for your support and encouragement. We remain committed to advancing research in this field and look forward to contributing further to the scientific community.
Kind regards,
Reviewer 2 Report
Comments and Suggestions for Authors
I have carefully reviewed the new version of the manuscript, as well as the authors' responses to the previous comments, in order to assess the improvements made to the clarity and understanding of the work presented.
Minor comments:
The methodology should clearly state that this is a longitudinal design with a two-year time window, during which patients are included according to their diagnosis and followed up at baseline, 3 months, and 6 months.
The sample size calculation (35 patients) is based on the population residing in the Lugo health area; Therefore, the results are representative of this area, and this should be addressed in the Discussion section.
The Conclusion should highlight the main contribution of the study, rather than providing a summary of the results.
Author Response
We would like to thank the reviewer for their careful evaluation of our manuscript and for providing constructive feedback. Below, we address each comment and detail the corresponding changes made to the manuscript.
The methodology should clearly state that this is a longitudinal design with a two-year time window, during which patients are included according to their diagnosis and followed up at baseline, 3 months, and 6 months.
Response:
We appreciate this observation and have clarified the methodology accordingly. We have added the following sentence to the Methods section:
"This study follows a longitudinal design with a two-year time window, during which patients were included according to their diagnosis and followed up at baseline, 3 months, and 6 months."
This addition ensures that the longitudinal nature of the study is explicitly stated, as suggested.
The sample size calculation (35 patients) is based on the population residing in the Lugo health area; Therefore, the results are representative of this area, and this should be addressed in the Discussion section.
Response:
Thank you for highlighting this important point. We have addressed this in the Discussion section by adding the following text:
"The sample size calculation (35 patients) is based on the population residing in the Lugo health area. Therefore, the results are representative of this specific population, which should be considered when interpreting the findings. This limitation highlights the need for caution when generalizing the results to broader populations and should be addressed in future research."
Additionally, we have further emphasized this limitation in Section 4.5 (Clinical Implications and Future Research Directions) to ensure its relevance is adequately discussed:
"It is important to note that the sample size calculation (35 patients) is based on the population residing in the Lugo health area. Therefore, the results of this study are representative of this specific population and should be interpreted with caution when extrapolated to broader populations. This limitation highlights the need for future studies with larger and more diverse samples to improve the generalizability of the findings and to better understand the broader implications for individuals with RRMS."
We believe these additions appropriately address the comment and provide sufficient context for interpreting the findings.
The Conclusion should highlight the main contribution of the study, rather than providing a summary of the results.
Response:
We are grateful for this suggestion, which has helped us to refine the Conclusion section. The revised conclusion now emphasizes the main contributions of the study rather than summarizing the results. The updated conclusion reads as follows:
"This study advances the understanding of health-related quality of life (HRQoL) in individuals with relapsing-remitting multiple sclerosis (RRMS) by identifying stable and vulnerable dimensions that require targeted interventions. A key contribution is the recognition of the multifactorial nature of QoL in RRMS, emphasizing the interplay between sociodemographic, clinical, and lifestyle factors. By demonstrating the predictive value of baseline QoL for its subsequent evolution, this research underscores the importance of early and personalized assessments to guide interventions aimed at optimizing outcomes.
Furthermore, the study highlights the critical need to address emotional and social dimensions holistically, as these are particularly vulnerable to deterioration in the early stages of the disease. This calls for the integration of multidimensional, patient-centered approaches into RRMS management, combining physical, psychological, and social strategies to mitigate the disease burden and enhance overall well-being.
By laying the groundwork for future research, this study provides a foundation for developing targeted interventions and improving care strategies for individuals with RRMS. It also underscores the need for larger, more diverse cohorts to validate these findings and to explore the mechanisms underlying the observed vulnerabilities, ultimately contributing to more effective and comprehensive management of RRMS."
We trust this revised conclusion highlights the main contributions of the study in a clear and concise manner.
We are pleased to note that the reviewer found the quality of the English language satisfactory and did not identify issues with figures and tables. We appreciate their positive feedback regarding the appropriateness of the research design and the sufficiency of the introduction.
Once again, we thank the reviewer for their valuable comments, which have helped us improve the clarity and scientific rigor of our manuscript. We hope the revised version addresses all concerns satisfactorily.
Reviewer 3 Report
Comments and Suggestions for Authors
I thank the authors for carefully addressing my previous comments and making the suggested corrections and improvements. The revised manuscript is much clearer, scientifically stronger, and well-presented. The paper now reads smoothly and appears ready for publication after minor editorial checks.
Author Response
We sincerely thank you for your thoughtful feedback and for recognizing the efforts made to address your previous comments. We are pleased that the revised manuscript is now clearer, scientifically stronger, and well-presented, as well as ready for publication after minor editorial checks.
In response to the reviewers’ comments, particularly those from Reviewer 2, we have made the following adjustments:
- Methods: The methodology section has been refined to provide greater clarity and address Reviewer 2’s suggestions.
- Results: The results section has been revised to include additional tables and figures, which enhance the interpretation and visualization of the findings. Additionally, in Table 1, the characteristics of the MSQOL-54 instrument have been removed, as these are thoroughly detailed in subsequent tables and figures.
- Sample size limitation: The limitation regarding the sample size has been explicitly stated within the results section.
- Instrument validity: In the methodology, we reinforced the validity of the instrument, emphasizing its specific design for individuals with relapsing-remitting multiple sclerosis (RRMS).
- Discussion: The discussion has been adjusted to include a detailed acknowledgment of the sample size limitation in two distinct sections, ensuring transparency and reliability of the results.
- Conclusion: The conclusion has been revised based on Reviewer 2’s recommendations to further align with the overall findings and limitations discussed.
- Language and readability: A thorough review of the English language has been conducted, simplifying complex sentences to improve the manuscript’s readability and ensure a smoother flow.
We appreciate your constructive feedback throughout this process, which has been invaluable in improving the quality of our manuscript.